# A microglia clonal inflammatory disorder in Alzheimer's disease

Rocio Vicario[1]*[†], Stamatina Fragkogianni[1†], Leslie Weber[1], Tomi Lazarov[1], Yang Hu[2], Samantha Y Hayashi[3], Barbara Craddock[3], Nicholas D Socci[4], Araitz Alberdi[1], Ann Baako[1], Oyku Ay[1], Masato Ogishi[5], Estibaliz Lopez-Rodrigo[1], Rajya Kappagantula[6], Agnes Viale[4], Christine A Iacobuzio-Donahue[6,7], Ting Zhou[8], Richard M Ransohoff[9], Richard Chesworth[9], Netherlands Brain Bank[10], Omar Abdel-Wahab[6], Bertrand Boisson[5], Olivier Elemento[2], Jean-Laurent Casanova[5], W Todd Miller[3], Frédéric Geissmann[1]*

[1]Immunology Program, Sloan Kettering Institute, Memorial Sloan Kettering Cancer Center, New York, New York, United States; [2]Department of Physiology and Biophysics, Institute for Computational Biomedicine, Weill Cornell New York, New York, United States; [3]Department of Physiology and Biophysics, Stony Brook University School of Medicine, Stony Brook, New York, United States; [4]Marie-Josée & Henry R. Kravis Center for Molecular Oncology, Memorial Sloan Kettering Cancer Center, New York, New York, United States; [5]St. Giles Laboratory of Human Genetics of Infectious Diseases, Rockefeller Branch, The Rockefeller University, New York, New York, United States; [6]Human Oncology & Pathogenesis Program, Memorial Sloan Kettering Cancer Center, New York, New York, United States; [7]Department of Pathology, Memorial Sloan Kettering Cancer Center, New York, New York, United States; [8]SKI Stem Cell Research Core, Memorial Sloan Kettering Cancer Center, New York, New York, United States; [9]Third Rock Ventures, Boston, United States; [10]Netherlands Brain Bank, Amsterdam, Netherlands

*For correspondence:
rociovicario@gmail.com (RV);
rociovicario@gmail.com (RV);
geissmaf@mskcc.org (FG)

[†]These authors contributed equally to this work

Competing interest: The authors declare that no competing interests exist.

## eLife assessment

This **fundamental** study enhances our understanding of how somatic variants in microglia might influence the onset and progression of neurodegenerative diseases such as Alzheimer's. The evidence supporting the conclusions is **compelling**, with the authors employing a multi-faceted approach to identify an enrichment of potentially pathogenic somatic mutations in Alzheimer's disease microglia. This research will be of significant interest to those investigating somatic mutations, Alzheimer's disease, microglial biology and cell signalling pathways.

**Abstract** Somatic genetic heterogeneity resulting from post-zygotic DNA mutations is widespread in human tissues and can cause diseases, however, few studies have investigated its role in neurodegenerative processes such as Alzheimer's disease (AD). Here, we report the selective enrichment of microglia clones carrying pathogenic variants, that are not present in neuronal, glia/stromal cells, or blood, from patients with AD in comparison to age-matched controls. Notably, microglia-specific AD-associated variants preferentially target the MAPK pathway, including recurrent CBL ring-domain mutations. These variants activate ERK and drive a microglia transcriptional program characterized by a strong neuro-inflammatory response, both in vitro and in patients. Although the natural history of AD-associated microglial clones is difficult to establish in humans, microglial expression of a MAPK pathway activating variant was previously shown to cause neurodegeneration

in mice, suggesting that AD-associated neuroinflammatory microglial clones may contribute to the neurodegenerative process in patients.

## Introduction

Neurodegenerative diseases are a frequent cause of progressive dementia. AD is diagnosed in ~90% of cases, with an estimated prevalence of ~10% in the population over 65 y of age (*Hebert et al., 2013*; *Alzheimer's Association, 2019*). The role of germline genetic variation in neurodegenerative diseases and AD has been studied intensely. Although autosomal dominant forms of AD due to rare germline variants with high penetrance account only for an estimated ~1% of cases (*Lanoiselée et al., 2017*; *Goate et al., 1991*; *Chartier-Harlin et al., 1991*; *Levy-Lahad et al., 1995a*; *Levy-Lahad et al., 1995b*; *Rogaev et al., 1995*), a number of common variants were also shown to contribute to disease risk. Carriers of one germline copy of the epsilon4 (E4) allele of the apolipoprotein E gene (APOE4), present in ~15 to 20% of the population, have a threefold higher risk of AD, while two copies (~2 to 3% of the population) increase the risk by ~10 fold (*Saunders et al., 1993*; *Murrell et al., 2006*; *Sando et al., 2008*; *Lumsden et al., 2020*). Genome-wide association studies (GWAS) have identified an additional ~50 common germline variants that more moderately increase the risk of AD, including TREM2, CD33, and MS4A6A variants (*Jonsson et al., 2013*; *Guerreiro et al., 2013*; *McQuade and Blurton-Jones, 2019*). Interestingly, the APOE4 allele is responsible for an increased inflammatory and neurotoxic response of microglia and astrocytes in the brain of carriers (*Arnaud et al., 2022*; *Serrano-Pozo et al., 2021*; *Rodriguez et al., 2014*), and it was noted that the majority of the other germline AD-risk variants are located within or near genes expressed in microglia *McQuade and Blurton-Jones, 2019* and in particular at microglia-specific enhancers (*Nott et al., 2019*). These data, together with transcriptional studies (*Krasemann et al., 2017*; *Mathys et al., 2019*; *Keren-Shaul et al., 2017*) support the hypothesis that genetic variation in microglia may contribute to the pathogenesis of neurodegeneration and AD.

Somatic genetic heterogeneity (mosaicism), resulting from post-zygotic DNA mutations, is widespread in human tissues, and a cause of tumoral, developmental, and immune diseases (*Miller et al., 2021*; *Martincorena et al., 2015*; *Martincorena and Campbell, 2015*; *Behjati et al., 2014*). Additionally, the role of somatic variants in neuropsychiatric disorders is also suspected (*McConnell et al., 2017*). Mosaicism has been documented in the brain tissue of AD patients in several deep-sequencing studies (*Keogh et al., 2018*; *Wei et al., 2019*; *Park et al., 2019*), showing that the enrichment of putative pathogenic somatic mutations in the PI3K-AKT, MAPK, and AMPK pathway do occur in the brain of patients in comparison to controls (*Park et al., 2019*). However, these studies performed in whole brain tissue lacked cellular resolution and mechanistic insights, and the role of somatic mutants in neurodegenerative diseases remains poorly understood (*Miller et al., 2021*). Somatic variants that activate the PI3K-AKT-mTOR or MAPK pathways in neural progenitors are a cause of cortical dysplasia and epilepsy *D'Gama et al., 2017*; *Khoshkhoo et al., 2023*; *Koh et al., 2018*; *Lim et al., 2015* and developmental brain malformations (*Poduri et al., 2012*), while somatic variants that activate the MAPK pathway in brain endothelial cells are associated with arteriovenous malformations (*Nikolaev et al., 2018*). Interestingly, we reported that expression of a somatic variant activating the MAPK pathway in microglia causes neurodegeneration in mice (*Mass et al., 2017*), but the presence and contribution of microglial somatic clones in neurodegenerative diseases and AD remains unknown.

Here, we investigated the presence and nature of somatic variants in brain cells from control and AD patients. In an attempt to examine all brain cells at the same resolution, nuclei from neurons, glia cells, and microglia, which only represent ~5% of brain cells, were pre-sorted. Human microglia are reported to develop in embryos and renew by local proliferation within the brain (*Askew et al., 2017*; *Réu et al., 2017*; *Bian et al., 2020*). However, bone marrow-derived myeloid cells can enter the brain, in particular during pathological processes, and may not be distinguishable from resident microglia by transcriptomics alone *Kim, 2023*. In order to distinguish somatic variants carried by resident microglia from the ones carried by myeloid cells of peripheral origin, we analyzed matched peripheral blood from control and patients, to 'barcode' somatic mutants shared between microglia and blood. Finally, in order to achieve high sensitivity in the detection of variants that confer a proliferative or activation advantage (pathogenic mutations) and support the emergence or pathogenicity of mosaic clones (*Frank, 2010*), and/or that have been previously associated with neurological diseases, we initially

**eLife digest** Around 10% of people aged over 65 are estimated to have Alzheimer's disease. This progressive neurodegenerative condition leads to death of brain cells, memory loss, confusion and other life-altering symptoms.

Somatic mutations are changes in the genetic information of a cell other than sperm or eggs, which can result in alterations in gene function. As the mutant cells multiply, they form clones that also carry these changes – potentially resulting in groups of cells that behave differently from those in which those mutations are absent. Despite their importance, the role of somatic mutations in Alzheimer's disease remains poorly understood.

To investigate this question, Vicario, Fragkogianni, Weber, Lazarov et al. examined the genetic material of brain and blood cells obtained from individuals who had died either of Alzheimer's disease, or of other causes. The team focused their analysis on around 700 genes previously associated with neurodegenerative conditions. The results showed that, compared to individuals whose death was not due to neurological illnesses, harmful variants of those genes were present in higher numbers in the microglia cells of around 25% of Alzheimer's patients in their series. No such increase was detected in other blood or brain cell populations, regardless of the individuals' cause of death.

Microglia are cells tasked with helping to repair damage and fight off infections in the brain. Many of the harmful gene variants found in this population switched on a cell pathway known as the MAP Kinase pathway, which activated the cells and caused them to multiply. This, in turn, led to inflammation and may contribute to the death of neurons.

Together these findings indicate that developing a new class of therapeutics that inhibits the MAP Kinase pathway in microglia may help prevent irreversible brain damage in some patients with Alzheimer's disease.

performed a targeted deep-sequencing of a panel of 716 genes covering somatic variants reported in clonal proliferative disorders and genes associated with neurodegenerative diseases diseases.

We found that microglia from AD patients were enriched for pathogenic variants in comparison to age-matched controls. Furthermore, we found that these microglia-specific AD-associated variants preferentially target the MAPK pathway, including recurrent CBL ring-domain mutations. In addition, we showed that these variants drive a microglia transcriptional program characterized by a strong neuro-inflammatory response previously associated with neurotoxicity, including the production of IL1 and TNF, both in in vitro microglia models and in patients. The natural history of the AD-associated microglia clonal inflammatory disorder we describe here is difficult to establish. Specifically, we do not know whether it contributes to the onset of the neuro-inflammatory process at an early stage of the disease, or if microglia carrying pathogenic mutations preferentially expand later during the course of the disease in response to tissue inflammation. Under both hypotheses, however, the presence of neuro-inflammatory microglial clones may contribute to the neurodegenerative process in a subset of AD patients. This report reveals a previously unrecognized presence of AD-associated microglia harboring pathogenic somatic variants in humans and provides mechanistic insight for neurodegenerative diseases by delineating cell-type specific variant recurrence.

## Results

### Clonal diversity among brain cells and blood from controls and AD patients

We examined post-mortem frozen brain samples and matching blood from 45 patients with intermediate-onset sporadic AD and 44 control individuals who died of other causes, including 27 donors age and sex-matched donors with the AD cohort (*Figure 1A*; *Figure 1—figure supplement 1A* and *Supplementary file 1*). APOE risk allele frequency for patients and controls was comparable to published series (*Murrell et al., 2006*; *Sando et al., 2008*; *Lumsden et al., 2020*; *Figure 1—figure supplement 1A*), and analysis of germline mutations did not identify deleterious variants in the 140 genes associated with neurological diseases. Myeloid/microglia, neurons, and glia/stromal cells were purified by flow cytometry using antibodies against PU.1 and NeuN (*Evrony et al., 2012*; *Figure 1B*,

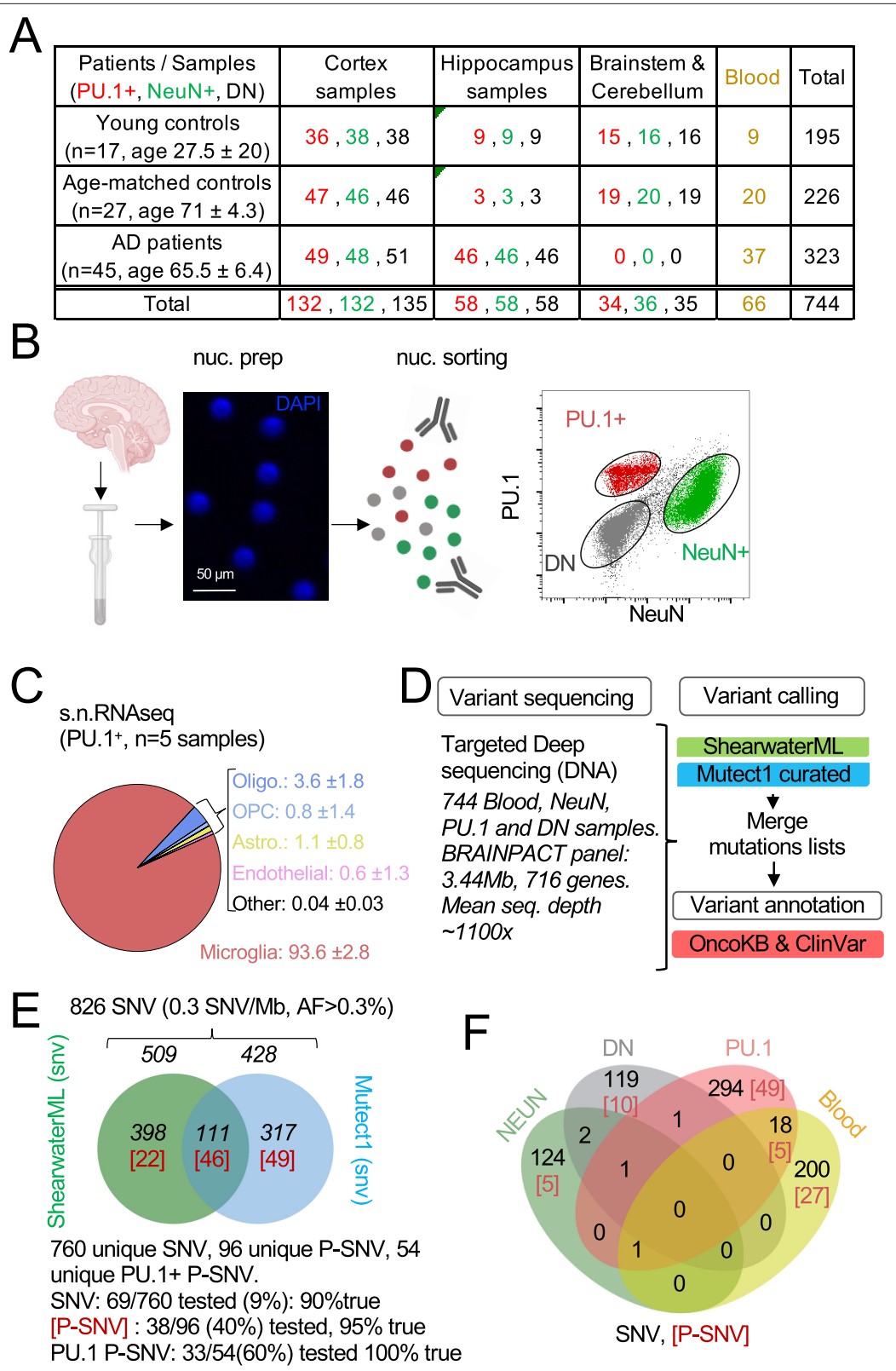

**Figure 1.** Detection of mutations in brain cell types and blood. (**A**) Table with patient and sample information. (**B**) Schematic represents the isolation and labeling of nuclei from post-mortem frozen brain samples from controls and Alzheimer's disease patients with DAPI and antibodies against PU.1+ (myeloid/microglia) and NeuN+ (neurons). Representative flow cytometry dot-plot of nuclei separation. Double negative nuclei are labeled 'DN.'

*Figure 1 continued on next page*

*Figure 1 continued*

(**C**) Percentage of cell types obtained in sorted PU.1⁺ nuclei determined by single-nuclei RNAseq in five brain samples from four individuals. (**D**) Schematic represents the sequencing strategy. Two algorithms (ShearwaterML and Mutect1) were used for variant calling. After annotation, pathogenicity was determined using OncoKb and ClinVar. (**E**) Venn diagram represents the number of variants and overlap between the ShearwaterML and Mutect1. Numbers in red indicate pathogenic variants (P-SNV). Validation of variants was performed by droplet digital (dd) PCR on pre-amplified DNA when available. (**F**) Venn diagrams represent the repartition per cell type of the 826 single-nucleotide variations (SNVs) identified in NeuN⁺: Neurons, PU.1⁺: microglia, DN: glia, and matching blood. [Numbers] in red indicate pathogenic variants P-SNV.

The online version of this article includes the following source data and figure supplement(s) for figure 1:

**Source data 1.** Source data for panel 1C.

**Figure supplement 1.** Quality control for DNA analysis and snRNA-seq.

**Figure supplement 1—source data 1.** Source data corresponding to panels A, B, C, and I.

*Figure 1—figure supplement 1B and C*). Single nuclei (sn)RNA-seq was performed on PU.1⁺ nuclei from one control and three AD patients to evaluate microglia enrichment following PU.1⁺ purification, and a cell-type annotation analysis indicated that ~94% of PU.1⁺ nuclei correspond to microglia (*Figure 1C* and *Figure 1—figure supplement 1D–1H*). Cortex samples were obtained from all donors but hippocampus samples were mostly obtained from AD patients (*Figure 1A*; *Supplementary file 1*). A total of 744 DNA samples from blood, PU.1⁺ nuclei, NeuN⁺ nuclei, and Double Negative nuclei (glia/stromal cells) from patients and controls (*Figure 1A*) were submitted to targeted hybridization/capture and deep-DNA targeted sequencing (TDS, *Figure 1D*, see Materials and methods), at mean coverage of ~1100 x (*Figure 1—figure supplement 1I*), for a panel of 716 genes (3.43 Mb, referred to below as BRAIN-PACT) which included genes reported to carry somatic variants in clonal proliferative disorders (n=576 genes) *Cheng et al., 2015*; *Durham et al., 2019* or that have been reported to be associated with neurodegenerativediseases (n=140) (*Bras et al., 2012*; *Renton et al., 2014*; *Karch et al., 2014*; *Karch and Goate, 2015*; *Turner et al., 2013*; *Ferrari et al., 2015*; *Kouri et al., 2015*; *Scholz and Bras, 2015*; *Nalls et al., 2014*; *Supplementary file 2*, see Materials and methods).

After QC and filtering of germline variants, variant calling using ShearwaterML, and a curated Mutect1 analysis identified 826 somatic synonymous and non-synonymous single-nucleotide-variations (SNVs), at an allelic frequency >0.3% (mean 1.3%) in the 744 samples, corresponding to an overall variant burden of 0.3 mut/Mb (*Figure 1E*). Sixty-six/826 SNV were present in more than one sample (*Supplementary file 3*). Droplet digital-PCR performed on pre-amplification DNA for ~10% of the 760 unique SNV was positive in 90% of cases (*Figure 1E*; *Supplementary file 3*). After annotation using the OncoKB *Chakravarty et al., 2017* and ClinVar (*Landrum et al., 2014*) databases for disease-associated or causative variants (*Figure 2D and F*; *Supplementary file 3*), 96 unique SNV were classified as Pathogenic (P)-SNV. 40% of these P-SNV were tested by droplet digital-PCR and confirmed in 95% of cases (*Figure 1E* and *Supplementary file 3*). Positive and negative results in matching brain samples from individual donors were confirmed in 100% of samples at a mean depth of ~5000 x (range 648–23.000 x) (*Supplementary file 3*). A venn-diagram analysis of SNVs detected in PU.1⁺, NeuN⁺, DN, and blood samples indicated that most (>90%) SNV and P-SNV were cell-type or tissue-specific, with ~5% of SNV and ~8% of P-SNV shared between the blood and brain of individual donors (*Figure 1F*; *Supplementary file 3*). These data indicate that targeted deep-sequencing of purified nuclei allows to detect of clonal mosaic variants with high sensitivity and specificity. In addition, 'barcoding' of clonal variants across tissues suggests that infiltrating myeloid cells of peripheral origin account for ~5% of microglia somatic diversity, and therefore that blood clones have a detectable but minor contribution to microglia, consistent with its local maintenance and proliferation (*Askew et al., 2017*; *Réu et al., 2017*).

Somatic clonal diversity of the different cell types, as evaluated by the SNV/megabase burden was higher in blood (1 mut/Mb) and PU.1⁺ nuclei (0.5 mut/Mb) than for DN and neurons (0.18 mut/Mb) (*Figure 2—figure supplement 1A*). The SNV/mb burden of blood and PU.1⁺ nuclei increased as a function of age (*Figure 2A* and *Supplementary file 3*) as previously reported for proliferating cells (*Martincorena et al., 2015*; *Martincorena et al., 2018*; *Jaiswal et al., 2014*; *Genovese et al., 2014*). Interestingly, the SNV/mb burden of blood cells from age-matched controls was higher than for AD patients (*Figure 2B*). In contrast, there was no difference in SNV/mb burden between PU.1⁺, NEUN⁺,

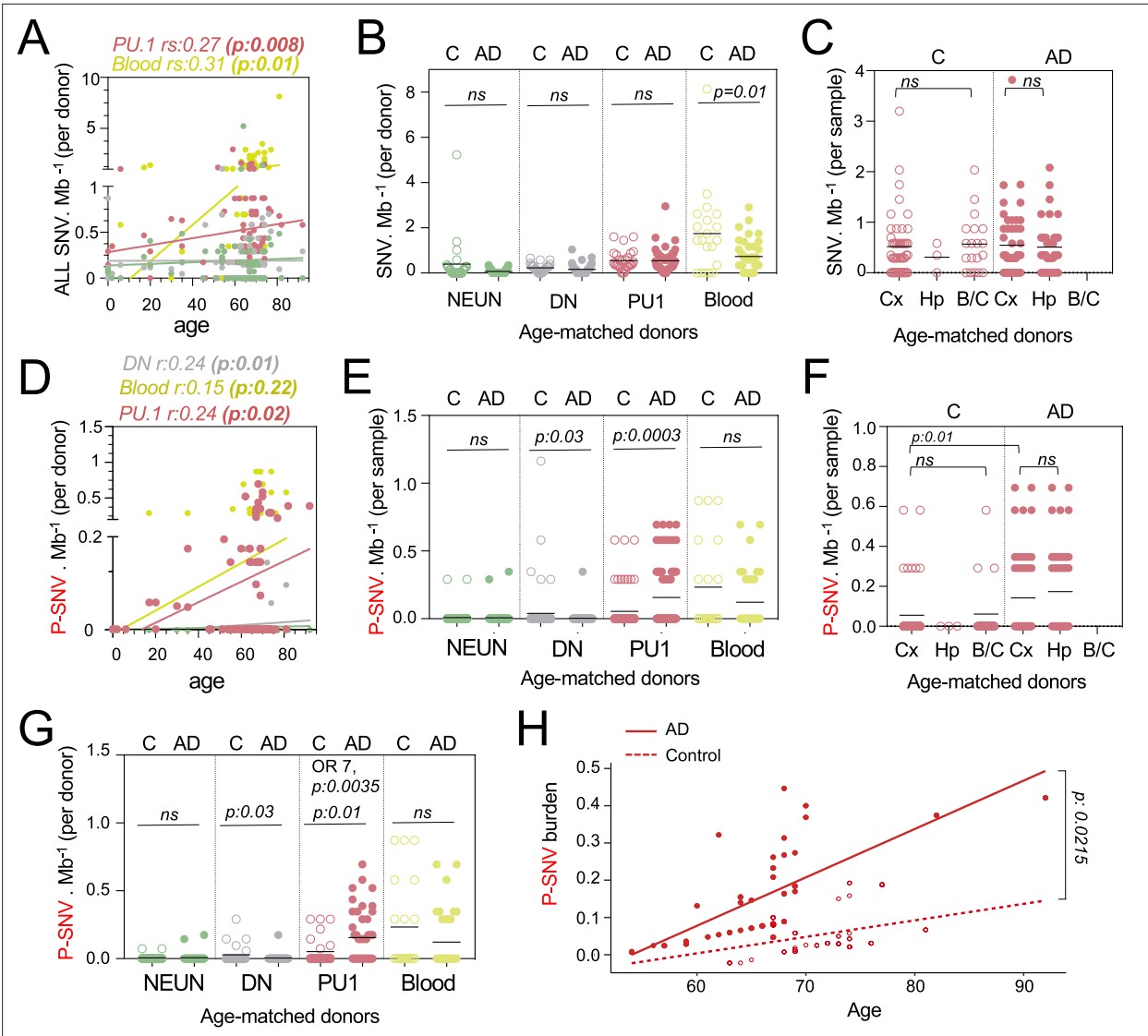

**Figure 2.** Pathogenic variants are enriched in microglia from Alzheimer's disease (AD) patients. (**A**) Correlation plot represents the mean number of variants per cell type and donor (n=89) (Y-axis), as a function of age (X-axis). Each dot represents mean value for a donor. Statistics: fitted lines, the correlation coefficients (rs), and associated p-*values* were obtained by linear regression (Spearman's correlation). (**B**) Number of single-nucleotide-variation (SNV) per Mb and cell types per donor, of age-matched controls (n=27) and AD patients (n=45). Each dot represents mean value for a donor. Statistics: *p-values* are calculated with unpaired two-tailed Mann-Whitney U. Note: non-parametric tests were used when data did not follow a normal distribution (D'Agostino-Pearson normality test). (**C**) Number of SNV per Mb in PU.1+ samples across brain regions, of age-matched controls (n=27) and AD patients (n=45). Each dot represents a sample. Statistics: *p-values* are calculated with Kruskal–Wallis, multiple comparisons. Note: non-parametric tests were used when data did not follow a normal distribution (D'Agostino-Pearson normality test). (**D**) Correlation plot represents the mean number of pathogenic variants (P-SNV) as determined by ClinVar and/or OncoKB, per cell type and donor (n=89) (Y-axis), as a function of age (X-axis). Each dot represents mean value for a donor. Statistics: fitted lines, the correlation coefficients (rs) and associated *p values* were obtained by linear regression (Spearman's correlation). (**E**) Number of P-SNV per Mb and cell types per sample, of age-matched controls (n=27) and AD patients (n=45). Each dot represents a sample. Statistics: *p-values* are calculated with unpaired two-tailed Mann-Whitney U. Note: non-parametric tests were used when data did not follow a normal distribution (D'Agostino-Pearson normality test). (**F**) Number of P-SNV per Mb in PU.1 samples across brain regions, of age-matched controls (n=27) and AD patients (n=45). Each dot represents a sample. Statistics: *p-values* for comparison within each group (controls and patients) are calculated with Kruskal–Wallis test and Dunn's test for multiple comparisons. *p-values* for the comparison of P-SNV between the cortex of C and the cortex of AD (0.01) was calculated with unpaired two-tailed Mann-Whitney U. Note: non-parametric tests were used when data did not follow a normal distribution (D'Agostino-Pearson normality test). (**G**) Number of P-SNV per Mb and cell types per donor for age-matched controls (n=27) and AD patients (n=45). Each dot represents mean value for a donor. Statistics: *p-values* are calculated with unpaired two-tailed Mann-Whitney U test. Odds ratio (95% CI, 2.049–29.02) and *p values* for the association between AD and the presence of pathogenic variants are calculated by multivariate logistic regression, with age and sex as covariates. (**H**) P-SNV burden as a function of age and disease status (age-matched controls and AD). Linear lines

*Figure 2 continued on next page*

*Figure 2 continued*

represent trend lines from mixed-effects linear regression that incorporates individual donor as a random effect (blue, control: p=0.0025, R^2=0.13; red, NDD: $P$=9.1 × 10^−16, R^2=0.50 by Pearson's correlation). The model's total explanatory power is substantial (conditional R^2=0.48). Both age and AD are associated with a significant increase in SNV burden in this model ($P$<1 × 10^−4 and $P$=1 × 10^−4, respectively, by likelihood ratio test). Anatomical regions of the brain specimen and originating brain banks were not incorporated because the models incorporating those parameters did not significantly improve the overall model fitting by likelihood ratio test (see Methods). Graph depicts SNV burden corrected by the mixed-effects model (See *Figure 2—figure supplement 1E* for observed P-SNV burden).

The online version of this article includes the following source data and figure supplement(s) for figure 2:

**Source data 1.** Data corresponding to panels A, B, C, D, E, F, G, and H.

**Figure supplement 1.** Analysis of pathogenic variants.

**Figure supplement 1—source data 1.** Data corresponding to panels A, B, C, D, E, and G.

**Figure supplement 2.** Summary of Alzheimer's disease (AD) patients characteristics and pathogenic variants.

and DN samples from AD patients and age-matched controls (*Figure 2B*), and between PU.1$^+$ nuclei from the cortex, hippocampus, and brainstem/cerebellum samples (*Figure 2C*). These data altogether indicate that the clonal diversity of microglia and blood both increase with age, and that the clonal diversity of blood cells is lower in AD than in age-matched controls who died of other causes including cancer and cardiovascular diseases (see Materials and methods). This is consistent with recent studies showing that clonal hematopoiesis is associated with a higher risk of several diseases related to ageing such as cardiovascular diseases, but is inversely associated with the risk of AD (*Jaiswal and Ebert, 2019*; *Bouzid et al., 2023*).

## Microglia clones carrying pathogenic variants are enriched in AD patients

In contrast to the global SNV burden, increased P-SNV burden was correlated not only with age (*Figure 2D*), but also with the disease status (AD) (*Figure 2E–H*). Within the control group, the SNV and P-SNV burden was higher in the blood of controls treated for cancer (*Figure 2—figure supplement 1B*). The P-SNV burden per Mb was selectively and highly enriched in PU.1$^+$ samples from AD patients in comparison to age-matched controls (p=0.0003, *Figure 2E*). Analysis of PU1$^+$ P-SNV/Mb burden per brains region indicated that the P-SNV/Mb burden was similar between brain regions within each group (*Figure 2F* and *Figure 2—figure supplement 1C*), and therefore attributable to AD status rather than sampling bias. Analysis of mutational load per donor confirmed that microglial clones carrying P-SNV were enriched in the brain of AD patients in comparison to age-matched controls (*Figure 2G*). Despite the relatively modest cohort size, a logistic regression analysis confirmed the association between the presence of P-SNVs in PU.1$^+$ nuclei and AD after adjusting for sex and age (*OR = 7*; p=0.0035, *Figure 2G* and *Figure 2—figure supplement 1D*). A mixed-effects linear regression model analysis also showed an excess of P-SNVs in AD independently of the effect of age (p=0.0215) (*Figure 2H* and *Figure 2—figure supplement 1E*).

In addition, genes targeted by P-SNV were all expressed in microglia (*Figure 2—figure supplement 1F* and *Supplementary file 3*) and the analysis of P-SNV/Mb mutational load restricted to genes that are not expressed in microglia did not show an enrichment of candidate pathogenic variants in AD patients (*Figure 2—figure supplement 1G* and *Supplementary file 3*). Altogether, these results show an association between microglia clones carrying P-SNV and AD in this series.

## AD patients carry microglial clones with MAP-Kinase pathway variants including recurrent CBL variants

Pathways analysis of genes carrying P-SNV in microglia from AD patients, against the background of the 716 genes sequenced, showed that the most significant pathways enriched were the receptor tyrosine kinase/MAP-Kinase pathways (Reactome, GO, and canonical pathways, *Figure 3A* and *Supplementary file 4*), corresponding to pathogenic/oncogenic variants in 6 of the 15 genes of the classical MAPK pathway (*Rauen, 2013*) (CBL, BRAF, RIT1, NF1, PTPN11, KRAS), TEK**,** and the KEGG Chronic Myeloid Leukemia (CML) pathway, which includes the former plus SMAD5 and TP53 (*Figure 3B, C*, *Figure 2—figure supplement 2*). Mutational load for MAPK genes was significantly higher in AD patients in comparison to age-matched control (*Figure 3C*). Other enriched pathways, albeit less

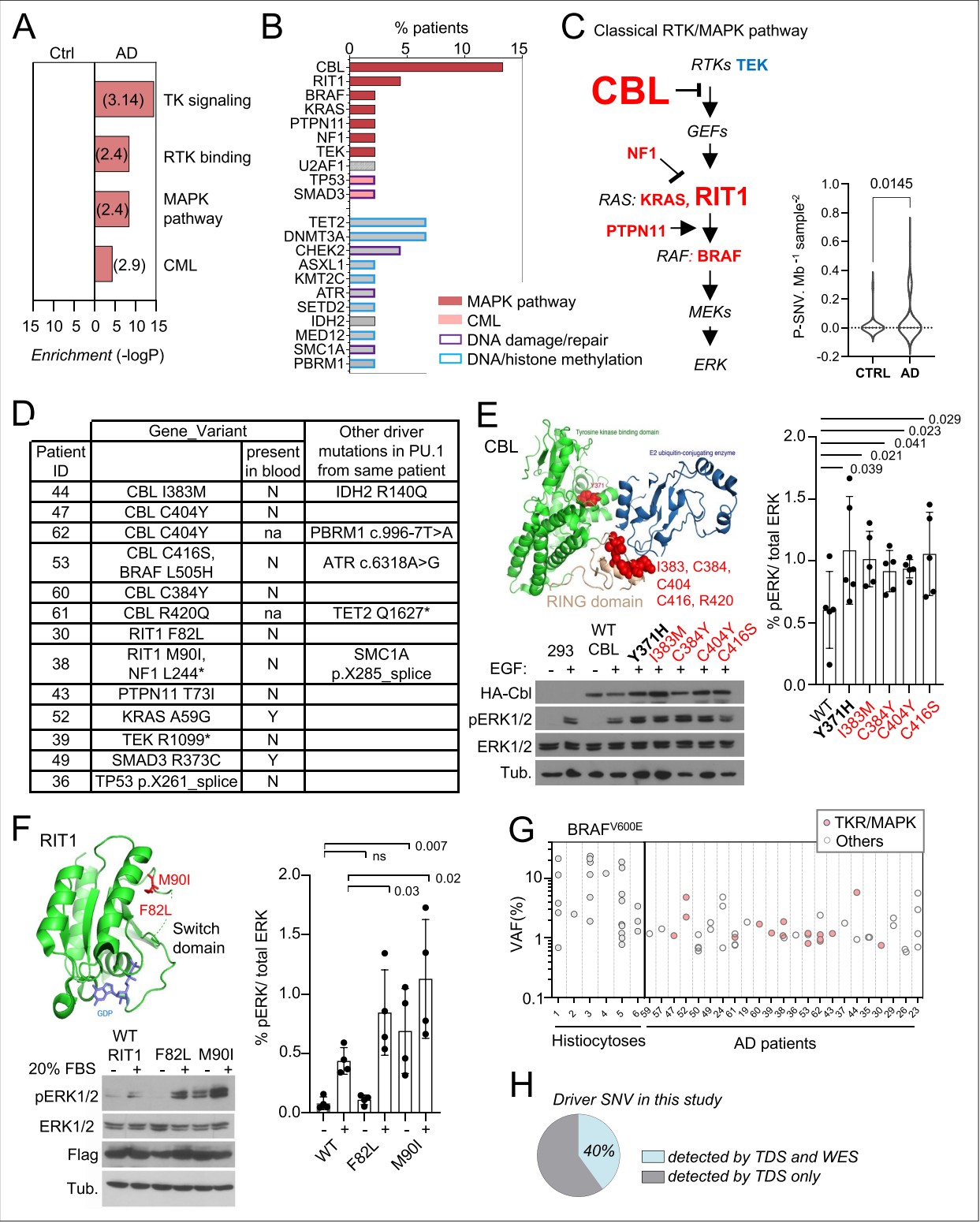

**Figure 3.** Somatic microglial clones with multiple and recurrent CBL and MAP-Kinase pathway activating variants. (**A**) Pathway enrichment analysis for the genes target of pathogenic variants (P-SNVs) using the panel of 716 genes as background set. Graph shows the most enriched pathways by: Reactome Gene Sets, GO Molecular Functions, Canonical Pathways and KEGG Pathway (see complete list in *Supplementary file 4*). (**B**) Bar plot indicates the genes carrying P-SNV (y-axis) and the % of Alzheimer's disease (AD) patients carrying P-SNV for each gene (x-axis). Genes are color-coded by pathway. (**C**) Representation of the classical MAPK pathway, the six genes mutated in AD patients are labeled in red, TEK is labeled in blue, and larger font size indicate reccurence of variants in a given gene. Violin plot shows distribution of P-SNV in genes from classical MAPK pathway per Mb

*Figure 3 continued on next page*

*Figure 3 continued*

sequenced and per sample in patients and controls, *p-value*: unpaired two-tailed Mann-Whitney U test. (**D**) Summary Table showing patients carrying P-SNV in the classical RTK/MAPK pathway and Chronic Myeloid Leukemia (CML)-associated genes (see *Supplementary file 3*) and indicating the detection of variants in blood, and their association with other variants in microglia. (**E**) Recurrent variants in the ring-like domain of CBL are indicated in red on the diagram structure of gene, above representative Western blot from cell lysates from HEK293T cells expressing WT, positive control (Y371H), or CBL variants alleles found in patients, and stimulated with EGF or control, probed with antibodies against Phospho-p44/42 MAPK (Erk 1/2, Thr202/Tyr204), total p44/42 MAPK (Erk1/2), HA-tag, and tubulin (BOTTOM). Histogram (RIGHT) represents quantification of the increase of the Phospho-ERK1/2/total ERK1/2 ratio in western blots in n=5 independent experiments, statistics: unpaired one-tailed t-test. (**F**) RIT1 M90I and F82L are represented on the 3D structure of the gene (pdb code: 4klz, F82 is within a segment whose structure was not resolved) and representative western blot from HEK293T cells expressing Flag-RIT1 (WT and mutants) and treated -/+ 20% FBS before harvesting. Lysates were probed with antibodies against Phospho-p44/42 MAPK (Erk 1/2, Thr202/Tyr204), total p44/42 MAPK (Erk1/2, (MAPK)), Flag, and tubulin. Histogram (RIGHT) represents quantification of the increase of the Phospho-ERK1/2/total ERK1/2 ratio in western blots in n=4 independent experiments, statistics: unpaired one-tailed t-test. (**G**) Variant allelic frequency (VAF, %) for the BRAF[V600E] allele in PU.1[+] nuclei from brain samples from histiocytosis patients (each dot represents a sample) and for P-SNVs in in PU.1+ nuclei from brain of AD patients (each dot represent a variant). Note: non-parametric tests were used when data did not follow a normal distribution (D'Agostino-Pearson normality test). (**H**) Percentage of P-SNVs detected by targeted deep sequencing (TDS) which were also detected by Whole-Exome-Sequencing (WES).

The online version of this article includes the following source data and figure supplement(s) for figure 3:

**Source data 1.** Data corresponding to panels A, B, C, E, F, and G.

**Source data 2.** Original PNG files for western blot analysis, indicating the relevant bands and treatments, displayed in panel E and F.

**Source data 3.** Unedited western blot JPEGs.

**Figure supplement 1.** Functional analysis of variants in HEK293 and BV2 cell lines.

**Figure supplement 1—source data 1.** Data corresponding to panel B.

**Figure supplement 1—source data 2.** Original PNG files for western blot analysis, indicating the relevant bands and treatments, displayed in panel A and B.

**Figure supplement 1—source data 3.** Unedited western blot JPEGs.

---

significant, included genes involved in DNA repair and chromatin binding/methyltransferase activity (*Figure 3B*; *Supplementary file 4*). No pathway was enriched in age-matched controls. Of note, we did not observe microglia P-SNVs within genes reported to be associated with neurological disorders (*Supplementary file 2*) in patients (*Supplementary file 3*). P-SNV targeting genes of the classical RTK/MAPK pathway (*Figure 3C*) were detected in the PU.1[+] samples from ~25% of the AD patients tested (p=0.0145 vs age-matched controls, *Figure 3D*, *Figure 2—figure supplement 2*). Strikingly, half of these patients (six patients, 13% of AD patients in this series) carried recurrent P-SNV in the RING domain of CBL (*Schnittger et al., 2012*; *Fernandes et al., 2010*; *Sargin et al., 2007*; *Dunbar et al., 2008*; *Bernard et al., 2014*; *Loh et al., 2009*; *Grand et al., 2009*; *Klampfl et al., 2013*; *Niemeyer et al., 2010*; *Javadi et al., 2013*; *Ogawa, 2019*; *Figure 3B–E*). Two additional patients presented with P-SNV in the Switch II domain of RIT1 (*Gómez-Seguí et al., 2013*; *Figure 3B–F*). Microglia from the three other patients carried activating KRAS (p.A59G), PTPN11 (p.T73I), and TEK (p.R1099*) oncogenic variants previously described in cancer and sporadic venous malformations (*Kim et al., 2016*; *Niihori et al., 2005*; *Soblet et al., 2013*; *Figure 3B and D*). In addition, a 12[th] patient carried a gain of function (GOF) U2AF1 (p.S34F) variant (*Okeyo-Owuor et al., 2015*), which is not a 'classical MAPK gene' but activates the MAPK pathway in myeloid malignancies (*Smith et al., 2019*; *Figure 3B*, *Figure 2—figure supplement 2*). Two patients carried two different MAPK activating variants: microglia from one patient carried an activating BRAF (p.L505H) variant *Choi et al., 2014* in addition to loss of function (LOF) variant CBL (p.C416S), and another patient carried the NF1 (p.L2442*) LOF variant *Heim et al., 1995*; *Bollag et al., 1996* in addition to the activating RIT1 (p.M90I) variant (*Figure 3D*, *Figure 2—figure supplement 2*). Five patients also carried additional P-SNV targeting genes involved in DNA repair with tumor suppressor function *Song et al., 2019*; *Zannini et al., 2014*, including the loss of function variants in ATR (c.6318A>G) *Fang et al., 2004* and SMC1A (p.X285_splice) (*Figure 3D*, *Figure 2—figure supplement 2*), and in DNA/histone methylation including TET2 (p.Q1627*) (*Schnittger et al., 2012*; *Haferlach et al., 2014*), IDH2 (p.R140Q) (*Ward et al., 2010*), and PBRM1 (c.996–7T>A) (*Brownlee et al., 2014*; *Figure 3D*, *Figure 2—figure supplement 2*). Finally, two patients carried oncogenic variants in genes from the KEGG Chronic Myeloid Leukemia (CML) pathway, SMAD3 (p.R373C) *Ku et al., 2007*, and TP53 (pX261_splice) *Bougeard et al., 2015* (*Figure 3D* and *Figure 2—figure supplement 2*). The detection of multiple

oncogenic variants in the same patients is reminiscent of the features observed in myeloproliferative disorders described outside the brain (*Ogawa, 2019*; *Haferlach et al., 2014*).

## Recurrent CBL and RIT1 variants activate the MAPK pathway

CBL is an E3 ubiquitin-protein ligase that negatively regulates RTK signaling via MAPK (*Liyasova et al., 2015*). CBL somatic and germ-line LOF variants such as R420Q have been previously associated with tumoral diseases including clonal myeloproliferative disorders *Schnittger et al., 2012*; *Fernandes et al., 2010*; *Sargin et al., 2007*; *Dunbar et al., 2008*; *Bernard et al., 2014*; *Loh et al., 2009*; *Grand et al., 2009*; *Klampfl et al., 2013*; *Niemeyer et al., 2010*; *Javadi et al., 2013*; *Ogawa, 2019* and RASopathies (*Brand et al., 2014*), respectively. We confirmed that CBL RING-domain variants found in AD patients increased MAPK phosphorylation in response to EGF upon expression of HA-tagged WT or mutant alleles in HEK293T cells (*Figure 3E*). RIT1 is a RAS GTPase, and somatic or germ-line GOF variants such as RIT1 F82L and RIT1 M90I, also enhance MAPK signaling in malignancies *Gómez-Seguí et al., 2013* and RASopathies (*Meyer Zum Büschenfelde et al., 2018*; *Aoki et al., 2013*), respectively. As in the case of CBL variants, the 2 RIT1 variants found in AD patients increased MAPK phosphorylation in response to FBS in HEK293T cells expressing these mutant alleles (*Figure 3F*). These data altogether indicate that a subset of AD patients (12/45, ~27% of this series) present with microglial clones carrying one or several oncogenic variants that activate the RTK/MAPK pathway, and are characterized by recurrent oncogenic variants in CBL and RIT1.

## Allelic frequency of the patients' MAPK activating variants

The allelic frequencies at which MAPK activating variants are detected in brain samples from AD patients range from ~1–6% in microglia (*Figure 3G*), which correspond to mutant clones representing 2 to 12% of all microglia in these samples, assuming heterozygosity. This range of allelic frequency is frequently observed for the MAPK-activating BRAF$^{V600E}$ variant in microglia isolated from brain samples of six patients diagnosed with *BRAF$^{V600E+}$* histiocytosis, a rare clonal myeloid disorder associated with neurodegeneration (*Mass et al., 2017*; *Boyd et al., 2020*; *Bhatia et al., 2020*; *Diamond et al., 2016*; *Héritier et al., 2018*; *Figure 3G* and *Supplementary file 5*). These data suggested that the size of the mutant microglial clones in AD patients was compatible with a role in a neuro-inflammatory/neuro-degeneration process.

## Other variants found in microglia from AD patients

Pathogenic variants that did not involve the MAPK pathway included LOF variants in the DNA repair gene CHEK2 including CHEK2 c.319+1 G>A *Cybulski et al., 2011* and CHEK2 R346H (*Figure 2—figure supplement 2* and *Figure 3—figure supplement 1A*), Mediator Complex gene MED12 (*Graham and Schwartz, 2013*), Histone methyltransferases SETD2 *Yang et al., 2016* and KMT2C/MLL3, the DNA methyltransferase DNMT3A (*Haferlach et al., 2014*; *Walter et al., 2011*), DNA demethylating enzymes TET2 and the Polycomb proteins ASXL1 (*Haferlach et al., 2014*). Of note, TET2, DNMT3, and KMT2C variants when present, were frequently detectable in the patients' matching blood at low allelic frequency (*Figure 2—figure supplement 2*). TET2, DNMT3, and KMT2C are frequently mutated in clonal hematopoiesis (*Jaiswal et al., 2014*; *Genovese et al., 2014*), suggesting that in contrast to other variants, the presence of TET2, DNMT3, and KMT2C/MLL3 in the brain of patients may reflect the entry of blood clones in the brain.

In half of the AD patients, no microglia pathogenic variants were identified. Targeted deep sequencing (TDS) cannot identify variants located outside of the BRAIN-PACT panel, such as other potential additional variants that would activate the MAPK pathway. Therefore, we performed whole exome sequencing (WES) of PU.1$^+$ nuclei at an average depth ~400 x, in selected samples from 48 donors, including samples from most of the patients negative for pathogenic variants by TDS (n=17 out of 22), a selection of patients with variants identified by TDS (n=16 out of 23), and 15 controls, followed by a curated Mutect analysis. Only 6/15 (40%) of the pathogenic SNVs previously identified by TDS and confirmed by ddPCR were detectable by WES in these samples (*Figure 3H*), indicating a lower sensitivity of WES. Nevertheless, after annotation by four modeling predictors Polyphen, SIFT, CADD/MSC, and FATHMM-XF (*Xi et al., 2004*; *Shihab et al., 2015*; *Adzhubei et al., 2010*; *Ng and Henikoff, 2001*; *Kircher et al., 2014*; *Itan et al., 2016*) additional SNVs predicted to be deleterious with high confidence were identified in 8/22 patients without pathogenic variants identified by

TDS (*Figure 2—figure supplement 2* and *Supplementary file 6*). Interestingly, four of the predicted deleterious variants identified by WES targeted genes that regulate the MAPK pathway ARHGAP9, ARHGEF26, CHD8, and DIXDC1 (*Figure 2—figure supplement 2* and *Supplementary file 6*).

## The patients' MAPK activating variants increase ERK phosphorylation, proliferation, inflammatory, and mTOR pathways in murine microglia and macrophages

CBL variants increased ERK phosphorylation upon lentiviral transduction in BV2 murine microglial cells (*Blasi et al., 1990*; *Henn et al., 2009*; *Figure 3—figure supplement 1B*). However, as this line was immortalized by v-Raf, which might interfere with the study of the MAPK pathway, we also stably expressed WT and variant CBL, RIT1, KRAS, PTPN11 alleles in SV-U19–5 transformed mouse 'MAC' lines (*Yu et al., 2008*; *Xiong et al., 2011*) (see Methods and *Figure 4—figure supplement 1A and B*). MAC lines expressing CBL, RIT1, KRAS, and PTPN11 variants presented with increased ERK phosphorylation and/or increontrols, as measured by Western immunoblotting and EdU incorporation (*Figure 4A* and *Figure 4—figure supplement 1A and B*). In addition, Hallmark and KEGG pathway analysis of RNAseq data from control and mutant lines showed increased RAS, TNF, IL6, and JAK STAT signaling, complement, inflammatory responses, and mTOR pathway activation signatures in mutants (*Figure 4B* and *Supplementary file 7*). These data indicated that microglia variants from patient's activate murine microglial cells and growth factor-dependent macrophages with proliferative and inflammatory responses in vitro. However, overexpression of mutant alleles in mouse cell lines does not necessarily recapitulate or predict the effects of a heterozygous genetic variant in physiological conditions. Thus, we investigated the role of CBL^C404Y allele in heterozygous human primary microglia-like cells.

## Heterozygosity for a CBL variant allele activates human microglia-like cells

We used prime editing *Anzalone et al., 2019* of human induced pluripotent stem cells (hiPSCs, see Methods) to generate isogenic hiPSCs clones heterozygous for the patients' variants (*Figure 4C* and *Figure 4—figure supplement 1*). We focused our analysis on CBL^404C/Y mutant lines because CBL was mutated in six patients and two of them carried the same *CBL c.1211G>A* p.C404Y variant (*Figure 2—figure supplement 2*). Microglia-like cells were differentiated from two independent hiPSC-derived CBL^404C/Y lines and their isogenic CBL^404C/C controls (*Figure 4C* and *Figure 4—figure supplement 1C*). CBL^404C/Y and isogenic CBL^404C/C microglia-like cells expressed similar amount of CBL total mRNA and protein, and CBL^404C/Y cells expressed wt and mutant mRNA in similar amounts, as expected assuming bi-allelic expression of CBL (*Figure 4—figure supplement 1D-5F*). CBL^404C/Y cells presented with a phenotype comparable to isogenic CBL^404C/C microglia-like cells for expression of IBA1, CSF1R, NGFR, EGFR, CD11b, MRC1, CD36, CD11c, Tim4, CD45, and MHC Class II (*Figure 4—figure supplement 1G*). Their viability was also comparable to control (*Supplementary file 5H*). However, CBL^404C/Y cells were larger and presented with more lamellipodia, resulting in an amoeboid morphology less frequently observed in isogenic controls (*Figure 4D and E*), and their proliferation rate was slightly increased, as measured by EdU incorporation (*Figure 4F*). Moreover CBL^404C/Y cells cultured in CSF1-supplemented medium also presented with a higher basal pERK level than control when restimulated with CSF1 (*Figure 4—figure supplement 1I*), and ERK phosphorylation after stimulation of starved microglia-like cells with CSF-1 was increased by ~ twofold in comparison to isogenic WT (*Figure 4G*). Altogether, these results showed that heterozygosity for a CBL^C404Y allele is sufficient to activate human microglia-like cells increasing their proliferation and ERK activation.

## Heterozygosity for a CBL^C404Y allele drives a microglial neuroinflammatory/AD associated signature

Gene Set Enrichment Analyses (GSEA) of RNAseq comparing CBL^404C/Y and isogenic CBL^404C/C microglia-like cells showed upregulation of Glycolysis, Oxidative Phosphorylation, and mTORC1 signatures, indicating increased metabolism and energy consumption by the mutant cells (*Figure 5A* and *Supplementary file 8*). In addition, as observed in MAC lines, CBL^404C/Y cells upregulated complement, TNF, and JAK STAT signaling and inflammatory signatures (*Figure 5A*; *Supplementary file 8*; *Ghosh et al., 2018*). Increased production of TNF, IL-6, IFN-γ, IL-1β, C3, and complement Factor

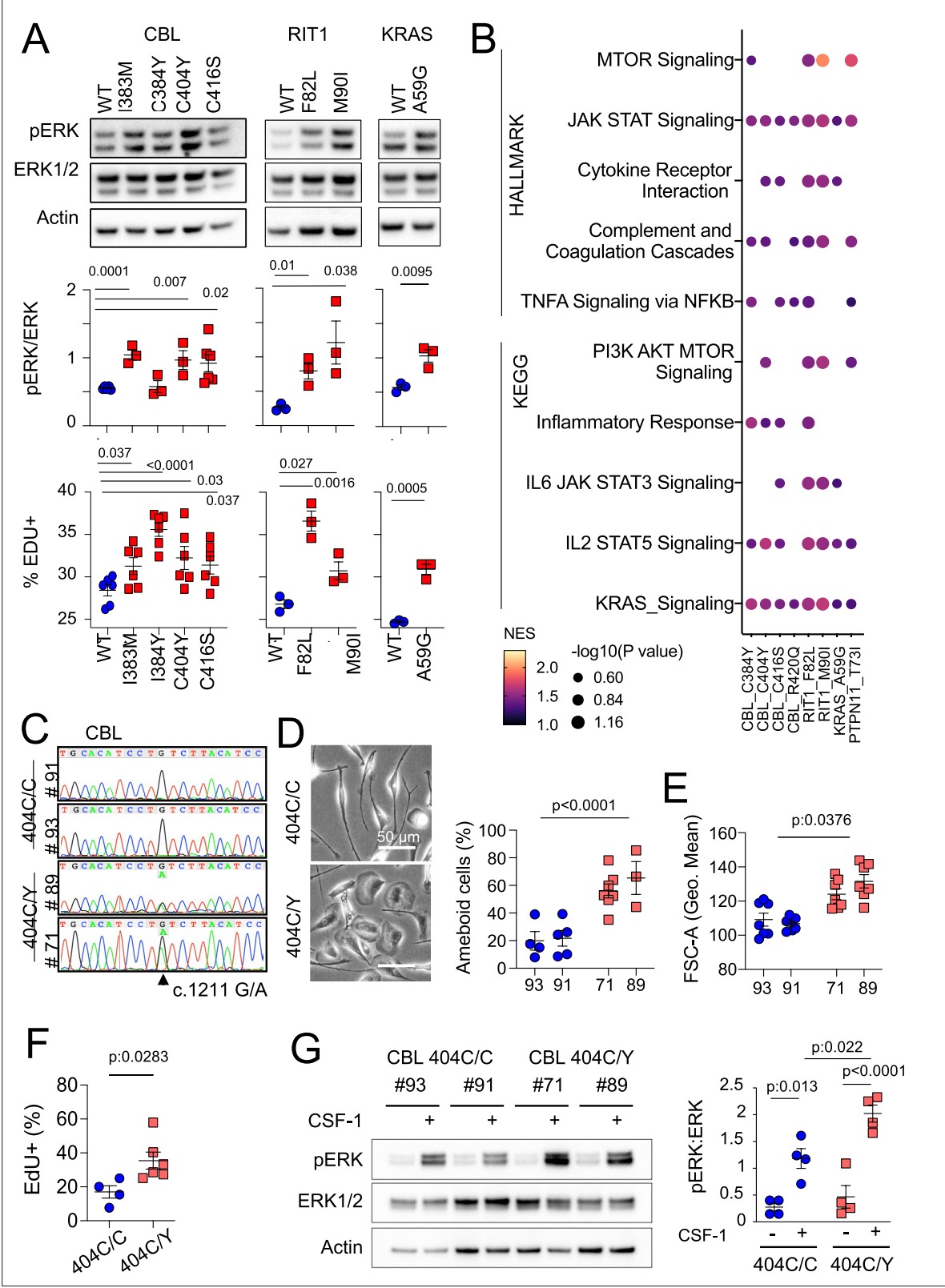

**Figure 4.** MAPK pathway activating variants in mouse macrophages and human induced Pluripotent Stem Cells (iPSC)-derived microglia-like cells. (**A**) Representative western-blot analysis (Top panels) and quantification (Middle panels) of phospho- and total-ERK in lysates from a murine CSF-1 dependent macrophage cell line expressing CBL$^{WT}$, CBL$^{I383M}$, CBL$^{C384Y}$, CBL$^{C404Y}$, CBL$^{C416S}$ (n=3–6), and RIT1$^{WT}$, RIT1$^{F82L}$ and RIT1$^{M90I}$ (n=3), KRAS$^{WT}$, and KRAS$^{A59G}$ (n=3). Bottom panels depicts flow cytometry analysis of EdU incorporation in the same lines. Statistics, Unpaired t-test. (**B**) HALLMARK and

*Figure 4 continued on next page*

*Figure 4 continued*

KEGG pathways (FDR/adj.p value <0.25, selected from *Supplementary file 7*) enriched in gene set enrichment analysis (GSEA) of RNAseq from mutant CSF-1 dependent macrophages lines CBL$^{I383M}$, CBL$^{C384Y}$, CBL$^{C404Y}$, CBL$^{C416S}$, CBL$^{R420Q}$, RIT1$^{F82L}$ RIT1$^{M90I}$, KRAS$^{A59G}$, and PTPN11$^{T73I}$ (n=3–6) in comparison with their wt controls. NES: normalized enrichment score. (**C**) Sanger sequencing of 2 independent hiPSC clones (#93 and #91) of CBL$^{404C/Y}$ heterozygous mutant carrying the c.1211G/A transition on one allele and 2 independent isogenic control CBL$^{404C/C}$ clones (#71 and #89) all obtained by prime editing. (**D**) Photomicrographs in CBL$^{404C/C}$ and CBL$^{404C/Y}$ iPSC-derived microglia-like cells.(**E**) Quantification of leading edge and lateral lamellipodia in CBL$^{404C/C}$ and CBL$^{404C/Y}$ iPSC-derived microglia-like cells. n=3–7, statistics: p-value are obtained by nested one-way ANOVA. (**F**) Flow cytometry analysis of cell size for the same lines (n>3) statistics: p-value are obtained with nested one-way ANOVA. (**G**) Flow cytometry analysis of EdU incorporation in CBL$^{404C/C}$ and CBL$^{404C/Y}$ microglia-like cells after a 2 hr EdU pulse. n=3, unpaired t-test. (**H**) Western-blot analysis (left) and quantification (right) of phospho- and total-ERK proteins in lysates from CBL$^{404C/C}$ and CBL$^{404C/Y}$ microglia-like cells starved of CSF-1 for 4 hr and stimulated with CSF-1 (5 min, 100 ng/mL) (n=4), statistics: p-value are obtained with two-way ANOVA.

The online version of this article includes the following source data and figure supplement(s) for figure 4:

**Source data 1.** Data corresponding to panels A, D, E, F, and G.

**Source data 2.** Original PNG files for western blot analysis, indicating the relevant bands and treatments, displayed in A and G.

**Source data 3.** Unedited western blot JPEGs.

**Figure supplement 1.** Analysis of mouse and human microglia-like cells.

**Figure supplement 1—source data 1.** Data corresponding to panels B, D, E, H, I.

**Figure supplement 1—source data 2.** Original PNG files for western blot analysis, indicating the relevant bands and treatments, displayed in panels A, B, F, and I.

**Figure supplement 1—source data 3.** Unedited western blot JPEGs.

H (CFH) by CBL$^{404C/Y}$ cells was confirmed by ELISA (*Figure 5B*). In addition, CBL$^{404C/Y}$ microglia-like cells also presented with signatures from the KEGG database associated with neurodegenerative disorders (*Figure 5A*; *Supplementary file 8*)**,** and for the recently published human microglia AD scRNA-seq signature, obtained by analysis of 24 sporadic AD patients and 24 controls (*Mathys et al., 2019*; *Figure 5C*). These data indicated that heterozygosity for the CBL$^{C404Y}$ allele is sufficient to drive expression of a neuroinflammatory/AD signature in a human microglia-like cell type, characterized by increased metabolism and the production of neurotoxic cytokines known to interfere with normal brain homeostasis.

## The MAPK variant neuroinflammatory microglial signature is detectable in patients

Analysis of the snRNA-seq data from five samples of purified microglia nuclei from four donors control, AD without and with pathogenic variants (*Figure 1C*, *Figure 1—figure supplement 1D–H* and *Supplementary file 9*) using unsupervised Louvain clustering and GSEA showed that microglia samples from patients carrying variants were enriched for the signatures observed in the MAC lines and CBL$^{404C/Y}$ cells (*Figure 5D and E* and *Figure 5—figure supplement 1*). In particular, microglia cluster 2 and 2B, were most enriched for the inflammatory, TNF, mTOR, and oxidative phosphorylation and glycolysis signatures in patients carrying variants (AD52, AD53) but not the controls (C11, AD34) (*Figure 5D and E* and *Figure 5—figure supplement 1*). Despite the small size of the mutant clones and the low sensitivity of scRNA-seq to detect rare allelic variants, KRAS A59G variant reads were detected in cluster 2/2B from patient AD52.

Altogether, the above results support the hypothesis that patients' microglial clones carrying pathogenic mutations are associated with a metabolic and neuroinflammatory signature that includes the production of neurotoxic cytokines in vitro and in vivo.

## Discussion

We report here that microglia from a cohort of 45 AD patients with intermediate-onset sporadic AD (mean age 65 y.o) is enriched for clones carrying pathogenic/oncogenic variants in genes associated with clonal proliferative disorders (*Supplementary file 2*) in comparison to 44 controls. Of note, we did not observe microglia P-SNVs within genes reported to be associated with neurological disorders in the patients.

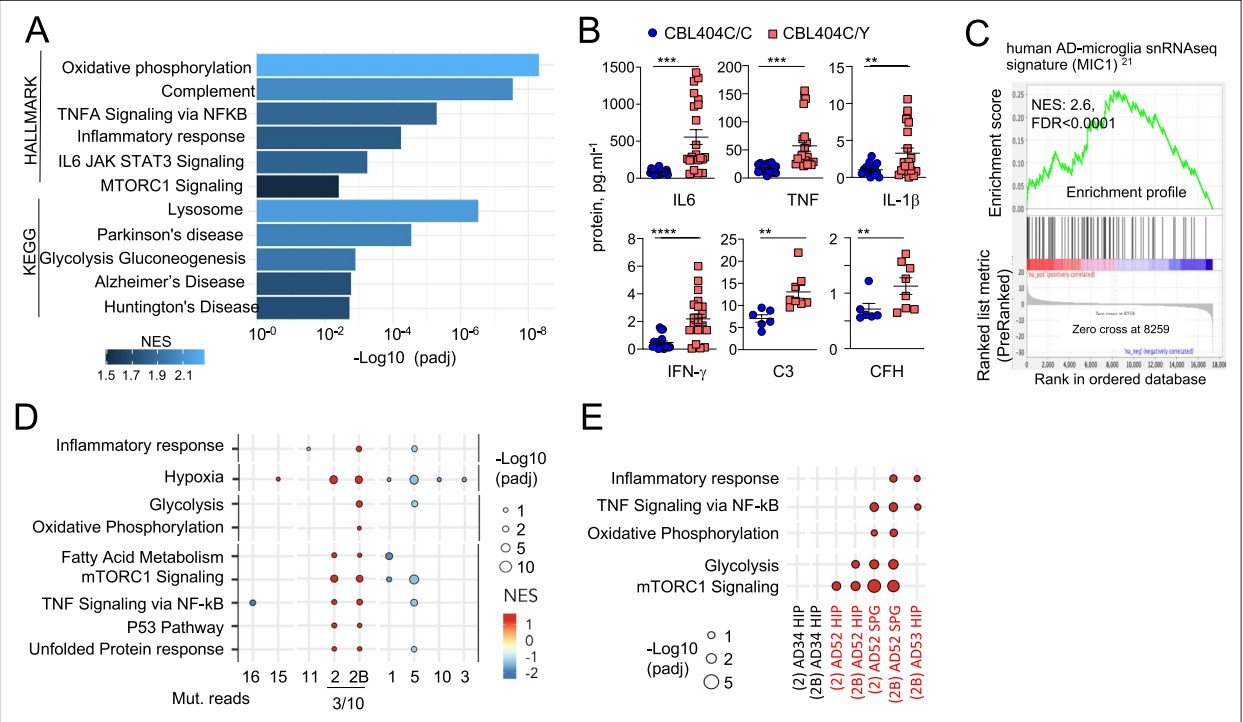

**Figure 5.** CBL[404C/Y] microglia signature. (**A**) HALLMARK and KEGG pathways (FDR /adj.p value <0.25, selected from *Supplementary file 8*) enriched in gene set enrichment analysis (GSEA) of RNAseq from from CBL[404C/Y] induced Pluripotent Stem Cells (iPSC)-derived macrophages and isogenic controls NES, normalized enrichment score. (**B**) ELISA for pro-inflammatory cytokines (n=3) and complement proteins (n=2) in the supernatant from CBL[404C/Y] iPSC-derived microglial-like cells and isogenic controls. Statistics: *p-value* are obtained by nonparametric Mann-Whitney U test,* *0.05, ** 0.01, *** 0.001, **** 0.0001*. (**C**) GSEA analysis for enrichment of the human AD-microglia snRNA-seq signature (MIC1) *Mathys et al., 2019* in differentially expressed genes between CBL[404Y/C] microglial-like cells and isogenic controls. (**D**) Dot plot represents the GSEA analysis of HALLMARK and KEGG pathways enriched in snRNAseq microglia clusters (samples from all donors). Genes are pre-ranked per cluster using differential expression analysis with SCANPY and the Wilcoxon rank-sum method. Statistical analyses were performed using the fgseaMultilevel function in fgsea R package for HALLMARK and KEGG pathways. Selected gene-sets with p-value <0.05 and adjusted p-value <0.25 are visualized using ggpubr and ggplot2 R package (gene sets/pathways are selected from *Figure 5—figure supplement 1B*, *Supplementary file 9*). (**E**) Dot plot represents the GSEA analysis (as in (**E**)) of HALLMARK and KEGG pathways enriched in cluster 2/2 B and deconvoluted by donor samples (selected from *Figure 5—figure supplement 1A*).

The online version of this article includes the following source data and figure supplement(s) for figure 5:

**Source data 1.** Data corresponding to panel B.

**Figure supplement 1.** snRNA-seq analysis of microglia.

These pathogenic variants are absent from blood, glia, or neurons in most cases. They are found predominantly in the MAPK pathway and include reccurrent variants (CBL RING domain variants in six patients), which promote microglial proliferation, activation, and expression of a neuroinflammatory/neurodegereration-associated transcriptional program in vitro and in vivo, and the production of neurotoxic cytokines IL1b, TNF, and IFNg (*Liu and Quan, 2018*; *Roy et al., 2020*; *Jayaraman et al., 2021*; *Ou et al., 2021*). Heterozygous expression of pathogenic CBL variant in human microglia-like cells was sufficient to drive a transcriptional program that associates with increased metabolic activity and a neurotoxic inflammatory response, also observed in microglia from patients with MAPK-activating variants.

The association between AD and MAPK pathway variants is consistent with a previous study where WES performed on unseparated brain tissue from AD patients showed that putative pathogenic somatic variants were enriched for the MAPK pathway, despite the lower sensitivity of the approach and the lack of cellular specificity (*Park et al., 2019*). The pathogenic role of the somatic pathogenic variants in the MAPK pathway associated with the microglia of AD patients is supported by several lines of evidence. We show here that they promote a neuroinflammatory/neurodegereration-associated transcriptional program in microglia-like cells. In addition, somatic variants that activate the MAPK pathway in tissue macrophages cause a clonal proliferative and inflammatory disease called

Histiocytosis, strongly associated with neurodegeneration (*Mass et al., 2017*; *Boyd et al., 2020*; *Bhatia et al., 2020*; *Diamond et al., 2016*), and introduction in mouse microglia of the variant allele most frequently associated with histiocytosis (BRAF$^{V600E}$) causes neurodegeneration in mice (*Mass et al., 2017*). The allelic frequencies of pathogenic variants found in AD patients are lower than values classically observed in solid tumors or leukemia, but within the range of the clonal frequency of pathogenic T cells observed in auto-immune diseases (*Thapa et al., 2015*), and we found that they were in the range of the allelic frequencies observed for the BRAF$^{V600E}$ variant in microglia in the brain of Histiocytosis patients. Moreover, the RAS/MAPK signaling pathway is involved in microglia proliferation, activation, and inflammatory response (*Lindberg et al., 2014*; *Qu et al., 2012*; *Coniglio et al., 2012*), neuronal death, neurodegeneration, and AD pathogenesis (*McQuade and Blurton-Jones, 2019*; *Nott et al., 2019*; *Mass et al., 2017*; *Scheltens et al., 2018*), and its activation has been proposed to be an early event in the pathophysiology of AD in human (*Lachén-Montes et al., 2016*). Neuroinflammation is an early event in AD pathogenesis, increasingly considered as critical in pathogenesis initiation and progression (*Arnaud et al., 2022*; *Schöll et al., 2015*; *Kinney et al., 2018*). This is underscored by the observation that the main known genetic risk factor for sporadic AD is the APOE4 allele, responsible for an increased inflammatory response in the brain of APOE4 carriers (*Arnaud et al., 2022*). In this regard, the contributing role of MAPK activating variants could be comparable to that of the APOE4 allele, and we noted that the allelic frequency of APOE4 allele is lower in patients with pathogenic variants (16/46 alleles, 34%) than in patients without detected variant (23/44 alleles, 53%) although the difference did not reach significance in this series.

Variants targeting the DNA-repair and DNA/histone methylation pathways are also enriched among AD patients, sometimes associated with the same patients, albeit their functional significance was not investigated here. Of note, however, germline variants of the DNA-repair transcription factor TP53, and DNA damage sensors ATR and CHEK2 were shown to promote accelerated neurodegeneration in human (*Song et al., 2019*; *Zannini et al., 2014*).

Microglia variants are frequently absent from blood, and our DNA sequencing barcoding approach does not support a model where blood cells massively infiltrate the brain or replace the microglia pool in patients from our series, but instead consistent with the local maintenance and proliferation of microglia (*Askew et al., 2017*; *Réu et al., 2017*). In addition, our results are consistent with a recent study showing that clonal hematopoiesis was inversely associated with the risk of AD (*Bouzid et al., 2023*).

The association of microglia clones carrying pathogenic variants with AD in a subset of patients is not a consequence of an overall increase in microglia mutational load (SNV) in AD. Together with evidence that pathogenic variants drive neuroinflammation, these data suggest that these clones could contribute to AD pathogenesis, together with other genetic and environmental factors. Lewy bodies, amyloid angiopathy, tauopathy, or alpha synucleinopathy, were equally distributed among AD patients with or without microglia clones carrying MAPK activating variants. The natural history of the microglial clones is difficult to study in human. It is possible that microglial clones with proliferative and activation advantages and a neuroinflammatory and neurotoxic profile may be present at the onset and contribute to the early stages of the disease. Alternatively, it is also possible that the microglial clones carrying the pathogenic mutations appear or are selected later during the course of the disease in the inflammatory milieu of the AD brain. In the latter case, pathogenic microglial clones may contribute to disease progression, i.e., neuroinflammation and neurodegeneration.

## Materials and methods

**Key resources table**

| Reagent type (species) or resource | Designation | Source or reference | Identifiers | Additional information |
|---|---|---|---|---|
| Cell line (*Homo sapiens*) | 293T cell line | ATCC | RRID:CVCL_0063 | |
| Cell line (*Mus musculus*) | BV2 microglial cell line | ARP American Research Products, Inc | RRID:CVCL_0182 | inmortalized with v-raf |

*Continued on next page*

*Continued*

| Reagent type (species) or resource | Designation | Source or reference | Identifiers | Additional information |
|---|---|---|---|---|
| Cell line (*Mus musculus*) | Mouse primary CSF-1 | gift of Dr. E. R. Stanley (Albert Einstein College of Medicine, Bronx, NY). | | immortalized with the SVU19-5 retrovirus |
| Cell line (*Homo sapiens*) | C12 induced Pluripotent Stem Cells (iPSCs) (female) | Derived from peripheral blood mononuclear cells (PBMCs) | | Used as a healthy wildtype control iPSC line (WT CBL) |
| Cell line (*Homo sapiens*) | 293T- Flag-tagged WT and 293 T-R346H CHK2 | This paper | | generated by transfection |
| Cell line (*Homo sapiens*) | 293T-Flag-tagged WT, 293T- RIT1 F82L, 293T-RIT1 M90I | This paper | | generated by transfection |
| Cell line (*Homo sapiens*) | 293T-CBL, 293T-CBLI383M, 293T-CBLC404Y, 293T-CBLC416S, 293T-CBLC384Y, 293T-CBLY371H | This paper | | generated by transfection |
| Cell line (*Mus musculus*) | BV2-empty vector, BV2-CBL, BV2-CBLI383M, BV2-CBLC404Y,BV2-CBLC416S, BV2-CBLC384Y, BV2-CBLY371H | This paper | | generated by viral transduction |
| Cell line (*Mus musculus*) | MAC-RIT1, MAC-RIT1 F82L, MAC-RIT1 M90I | This paper | | generated by viral transduction |
| Cell line (*Homo sapiens*) | MAC-CBL, MAC-CBLI383M, MAC-CBLC404Y, MAC-CBLC416S, MAC-CBLC384Y, MAC-CBLY371H | This paper | | generated by viral transduction |
| Cell line (*Homo sapiens*) | MAC-KRAS, MAC-KRASA59G | This paper | | generated by viral transduction |
| Cell line (*Homo sapiens*) | MAC-PTPN11, MAC- PTPN11T73I | This paper | | generated by viral transduction |
| Cell line (*Homo sapiens*) | IPSC-CBLC404Y | This paper | | C12 line genetically modified to contain CBLC404Y |
| Transfected construct (human) | Flag-tagged CHK2 | Sino Biological | | |
| Transfected construct (human) | Flag-tagged RIT1 | Origene | | |
| Transfected construct (human) | Flag-tagged RIT1M90I and Flag-tagged RIT1F82L | This paper | | generated by site-directed mutagenesis using the QuikChange Kit |
| Transfected construct (human) | pcDNA3-HA-tagged c-Cbl | gift from Dr. Nicholas Carpino (Stony Brook) | | |
| Transfected construct (human) | pcDNA3-HA-tagged- CBLI383M, CBLC404Y, CBLC416S, CBLC384Y, CBLY371H | This paper | | generated by site-directed mutagenesis using the QuikChange Kit |
| Transfected construct (human) | pHAGE_puro | gift from Christopher Vakoc | Addgene plasmid # 118692; http://n2t.net/addgene:118692; RRID:Addgene_118692 | |
| Transfected construct (human) | pHAGE-KRAS | gift from Gordon Mills & Kenneth Scott | Addgene plasmid # 116755; http://n2t.net/addgene:116755; RRID:Addgene_116755 | |
| Transfected construct (human) | pHAGE-PTPN11 | gift from Gordon Mills & Kenneth Scott | Addgene plasmid # 116782; http://n2t.net/addgene:116782; RRID:Addgene_116782 | |
| Transfected construct (human) | pHAGE-PTPN11-T73I | gift from Gordon Mills & Kenneth Scott | Addgene plasmid # 116647; http://n2t.net/addgene:116647; RRID:Addgene_116647 | |

*Continued on next page*

*Continued*

| Reagent type (species) or resource | Designation | Source or reference | Identifiers | Additional information |
|---|---|---|---|---|
| Transfected constructs (human) | Phage-CBL, Phage-CBLI383M, Phage-CBLC404Y, Phage-CBLC416S, Phage-RIT1, Phage-RIT1M90I, Phage-RIT1F82L, phage-KRASA59G and pHAGE-CBLC384Y | This paper | | generated by Azenta Life Sciences via a PCR cloning approach and targeted mutagenesis |
| Transfected construct (human) | pDONR223_KRAS_p.A59G | gift from Jesse Boehm & William Hahn & David Root | Addgene plasmid # 81662; http://n2t.net/addgene:81662; RRID:Addgene_81662 | |
| Biological samples (Homo-sapiens) | Biological samples (brain tissue and blood) from patients and controls | See 'Human sample collection and consent information' in methods for details | | |
| Biological samples (*Mus musculus*) | CF1 Mouse Embryonic Fibroblasts, irradiated | Thermo Fisher Scientific | A34181 | |
| Antibody | NeuN-PE. Mouse monoclonal | Milli-Mark | Cat# FCMAB317PE, RRID:AB_11212465 | used at 1:500 dilution |
| Antibody | Pu.1-AlexaFluor 647. Rabbit monoclonal | Cell Signaling | Cat# 2240, RRID:AB_2186911 | used at 1:50 dilution |
| Antibody | anti-Cdc42, rabbit polyclonal | Santa Cruz | Cat# sc-87, RRID:AB_631213 | 1 µg |
| Antibody | Phospho-p44/42 MAPK (Thr202/Tyr204) mouse monoclinal | Cell Signaling | Cat# 4370, RRID:AB_2315112 | used at 1:1000 dilution |
| Antibody | total p44/42 MAPK, rabbit polyclonal | Cell Signaling | Cat# 9102, RRID:AB_330744 | used at 1:1000 dilution |
| Antibody | HA, mouse monoclonal | Millipore | Cat# 05–904, RRID:AB_417380 | used at 1:1000 dilution |
| Antibody | Flag, mouse monoclonal | Sigma | Cat# A8592, RRID:AB_439702 | used at 1:1000 dilution |
| Antibody | pCHEK2 (T383), Rabbit Polyclona | Abcam | Cat# ab59408, RRID:AB_942224 | used at 1:500 dilution |
| Antibody | Anti-γ-Tubulin, mouse monoclonal | Sigma | Cat# T6557, RRID:AB_477584 | used at 1:10000 dilution |
| Antibody | anti-c-CBL, Rabbit Polyclonal | Cell Signaling | Cat# 2747, RRID:AB_2275284 | used at 1:1000 dilution |
| Antibody | anti-RIT1, Rabbit Polyclonal | Abcam | Cat# ab53720, RRID:AB_882379 | used at 1:1000 dilution |
| Antibody | anti-KRAS, Rabbit Polyclonal | Sigma | Cat# H00003845-M01, RRID:AB_540078 | 1 µg/mL |
| Antibody | anti-Actin, mouse monoclonal | Sigma | Cat# MAB1501, RRID:AB_2223041 | used at 1:10,000 dilution |
| Antibody | anti-rabbit IgG HRP-linked, goat polyclonal | Cell Signaling | Cat# 7074, RRID:AB_2099233 | used at 1:1000 dilution |
| Antibody | anti-mouse IgG HRP-linked, horse polyclonal | Cell Signaling | Cat# 7076, RRID:AB_330924 | used at 1:1000 dilution |
| Antibody | PE- anti-CD115 (CSF1-R), rat monoclonal | BD | Cat# 565368, RRID:AB_2739206 | used at 1:100 dilution |
| Antibody | PE/Cy7-conjugated anti-CD11b, mouse monoclonal | BD Biosciences | at# 557743, RRID:AB_396849 | used at 1:100 dilution |
| Antibody | Alexa Fluor 488-conjugated-anti-CD206, mouse monoclonal | Thermo Fisher Scientific | Cat# 53-2069-42, RRID:AB_2574416 | used at 1:100 dilution |
| Antibody | PE-conjugated anti-integrin, mouse monoclonal | R&D systems | Cat# FAB3050P, RRID:AB_920540 | used at 1:100 dilution |
| Antibody | PE/Cy5-conjugated anti-CD11c, mouse monoclonal | BD Biosciences | Cat# 551077, RRID:AB_394034 | used at 1:100 dilution |
| Antibody | APC-conjugated anti-Tim4, mouse monoclonal | Biolegend | Cat# 354007, RRID:AB_2564543 | used at 1:100 dilution |
| Antibody | PE/Cy7-conjugated anti-HLA-DR, mouse monoclonal | BD Biosciences | Cat# 560651, RRID:AB_1727528 | used at 1:100 dilution |
| Antibody | BV650-conjugated anti-CD45, mouse monoclonal | Thermo Fisher Scientific | Cat# 416-0459-42, RRID:AB_2925684 | used at 1:100 dilution |

*Continued on next page*

*Continued*

| Reagent type (species) or resource | Designation | Source or reference | Identifiers | Additional information |
|---|---|---|---|---|
| Antibody | APC/Cy7-conjugated anti-CD14, rat monoclonal | Biolegend | Cat# 123317, RRID:AB_10900813 | used at 1:100 dilution |
| Antibody | PE-conjugated anti-NGFR, mouse monoclonal | eBioscience | Cat# 12-9400-42, RRID:AB_2572710 | used at 1:100 dilution |
| Antibody | Alexa Fluor 647-conjugated anti-EGFR, mouse monoclonal | BD Pharmigen | Cat# 563577, RRID:AB_2738288 | used at 1:100 dilution |
| Antibody | APC/Cy7-conjugated anti- CD36, mouse monoclonal | Biolegend | Cat# 336213, RRID:AB_2072512 | used at 1:100 dilution |
| Antibody | APC-conjugated anti-CD172a (SIRPa), mouse monoclonal | Thermo Fisher Scientific | Cat# 17-1729-42, RRID:AB_1944409 | used at 1:100 dilution |
| Antibody | Alexa Fluor 555-conjugated anti-Iba1 antibody, Rabbit monoclonal | Cell Signaling | Cat# 36618, RRID:AB_2943227 | used at 1:100 dilution |
| Commercial assay or kit | KAPA Hyper Prep Kit | Kapa Biosystems | KK8504 | |
| Software, algorithm | muTect 1 | https://github.com/soccin/BIC-variants_pipeline and https://github.com/soccin/Variant-PostProcess; doi:10.1016/j.jmoldx.2014.12.006 | v1.1.7 | |
| Software, algorithm | ShearwaterML | Martincorena, I. et al. | | |
| Software, algorithm | FlowJo | BD | 10.6.2 | |
| Commercial assay or kit | QIAamp DNA Micro Kit | Qiagen | Cat#56304 | |
| Commercial assay or kit | HiSeq 3000/4000 SBS Kit | Illumina | | |
| Sequence-based reagent | KRAS_G12D, ddPCR | Bio-Rad | Unique Assay ID: dHsaMDV2510596 | |
| Sequence-based reagent | MTOR_Arg1616His_c.4847G>A | Bio-Rad | Unique Assay ID: dHsaMDV2510596 | |
| Commercial assay or kit | 10 X genomics Reagent Kit 3' v3.1 | 10 X genomics | | |
| Software, algorithm | Seurat v4.0.3 | https://github.com/satijalab; *Hao et al., 2021* | | |
| Commercial assay or kit | NovaSeq 6000 S4 Reagent Kit (200 Cycles) | Illumina | | |
| Commercial assay or kit | RNeasy Mini kit | Qiagen | | |
| Sequence-based reagent | CBL_I383M | Thermo Fisher Scientific | dHsaMDS675699482 | |
| Sequence-based reagent | CBL_C384Y | Thermo Fisher Scientific | dHsaMDS386449640 | |
| Sequence-based reagent | CBL_C404Y | Thermo Fisher Scientific | dHsaMDS437459772 | |
| Sequence-based reagent | CBL_mRNA_C404Y | Thermo Fisher Scientific | dMDS334857054 | |
| Sequence-based reagent | CBL_C416S | Thermo Fisher Scientific | dHsaMDS613275900 | |
| Sequence-based reagent | RIT1_ F82L | Thermo Fisher Scientific | dMDS959028273 | |
| Sequence-based reagent | RIT1_M90I | Thermo Fisher Scientific | dHsaMDS133045056 | |

*Continued on next page*

*Continued*

| Reagent type (species) or resource | Designation | Source or reference | Identifiers | Additional information |
|---|---|---|---|---|
| Sequence-based reagent | c-CBL FAM | Thermo Fisher Scientific | Hs01011446_m1 | |
| Sequence-based reagent | CBLb FAM | Thermo Fisher Scientific | Hs00180288_m1 | |
| Sequence-based reagent | GAPDH VIC | Thermo Fisher Scientific | Hs02786624_g1 | |
| Commercial assay or kit | Zombie Violet Viability | Biolegend | | |
| Commercial assay or kit | Cytofix/Cytoperm solution | BD Pharmingen | | |
| Commercial assay or kit | Click-iT Plus EdU Alexa Fluor 647 Flow Cytometry Assay Kit | Thermo Fisher Scientific | | |
| Commercial assay or kit | miRNeasy Mini Kit | Qiagen | | |
| Commercial assay or kit | MagMAX mirVana Total RNA Isolation Kit | Thermo Fisher Scientific | | |
| Commercial assay or kit | KingFisher Flex Magnetic Particle Processor | Thermo Fisher Scientific | | |
| Commercial assay or kit | TruSeq Stranded mRNA LT Kit | Illumina | | |
| Software, algorithm | R/Bioconductor package DESeq | EMBL Heidelberg | https://bioconductor.org/packages//2.10/bioc/html/DESeq.html | |

## Tissue samples

The study was conducted according to the Declaration of Helsinki. Human tissues were obtained with patient-informed consent and used under approval by the Institutional Review Boards from Memorial Sloan Kettering Cancer Center (IRB protocols #X19-027). Snap-frozen human brain and matched blood were provided by the Netherlands Brain Bank (NBB), the Human Brain Collection Core (HBCC, NIH), Hospital Sant Joan de Déu, and the Rapid Autopsy Program (MSKCC, IRB #15–021). Samples were neuropathologically evaluated and classified by the collaborating institutions as AD (*Dubois et al., 2007*; *Braak and Braak, 1991*; *Braak and Braak, 1995*; *McKhann et al., 1984*; *McKhann et al., 2011*) or non-dementia controls. The mean age of AD patients is 65 y old (55.5% female, 44.5% male). The mean age of all controls is 54 y old (60% female, 40% male), and the mean age of AD age-matched controls was 70 y old (60% female, 40% male). The overall mean of the post-mortem delay interval was 9.8 hr. Patients did not present with germline pathogenic PSEN1/2/3 or APP AD's associated variants. For additional information on donor's brain regions, sex, age, cause of death, Apoe status, Braak status see *Supplementary file 1*. To avoid possible contamination of sequencing data with mutations associated with donor's tumoral disease in the group of non-dementia controls, we refrained from selecting cases with blood malignancies or with brain tumors. Samples from histiocytosis patients were collected under GENE HISTIO study (approved by CNIL and CPP Ile de France) from Pitié-Salpêtrière Hospital and Hospital Trousseau and from Memorial Sloan Kettering Cancer Center.

## Nuclei isolation from frozen brain samples, FACS-sorting, and DNA extraction

All samples were handled and processed under Air Clean PCR Workstation. An average of 400 mg of frozen brain tissues were homogenized with a sterile Dounce tissue grinder using a sterile non-ionic surfactant-based buffer to isolate cell nuclei ('homogenization buffer:' 250 mM Sucrose, 25 mM KCL, 5 mM MgCl2, 10 mM Tris buffer pH 8.0, 0.1% (v/v) Triton X-100, 3 µM DAPI, Nuclease Free Water). Homogenate was filtered in a 40 µm cell strainer and centrifuged 800 g 8 min 4 °C. To clean-up the homogenate, we performed a iodixanol density gradient centrifugation as follows: pellet was gently mixed 1:1 with iodixanol medium at 50% (50% Iodixanol, 250 mM Sucrose, 150 mM KCL, 30 mM

MgCl2, 60 mM Tris buffer pH 8.0, Nuclease Free Water) and homogenization buffer. This solution layered to a new tube containing equal volume of iodixanol medium at 29% and centrifuged 13.500 g for 20 min at 4 °C. Nuclei pellet was gently resuspended in 200 µl of FACS buffer (0.5% BSA, 2 mM EDTA) and incubated on ice for 10 min. After centrifugation 800 g 5 min 4 °C, sample was incubated with anti-NeuN (neuronal marker, 1:500, Anti-NeuN-PE, clone A60 Milli-Mark) for 40 min. After centrifugation 800 g 5 min 4 °C, sample was washed with 1 X Permeabilization buffer (Foxp3 /Transcription Factor Staining Buffer Set, eBioscience) and centrifuged 1300 g for 5 min, without breaks to improve nuclei recovery. Staining with anti-Pu.1 antibody in 1 X Permeabilization buffer (myeloid marker 1:50, Pu.1-AlexaFluor 647, 9G7 Cell Signaling) was performed for 40 min. After a wash with FACS buffer samples were prepared for FACS. Nuclei were FACS-sorted in a BD FACS Aria with a 100 µm nozzle and a sheath pressure 20 psi, operating at ~1000 events per second. Nuclei were sorted into 1.5 ml certified RNAse, DNAse DNA, ATP, and Endotoxins tubes containing 100 µl of sterile PBS. For detailes on sorted samples see *Supplementary file 1*. Sorting purity was >95%. Sorting strategy is depicted in *Figure 1—figure supplement 1*. Of note, the Double-negative gate is restricted to prevent cross-contamination between cell types. Nuclei suspensions were centrifuged 20 min at 6000 g and processed immediately for gDNA extraction with QIAamp DNA Micro Kit (Qiagen) following manufacture instructions. DNA from whole-blood samples was extracted with QIAamp DNA Micro Kit (Qiagen) following manufacture instructions. Flow cytometry data was collected using DiVa 8.0.1 Software. Subsequent analysis was performed with FlowJo_10.6.2. For sorting strategy, see *Figure 1—figure supplement 1*.

## DNA library preparation and sequencing

DNA samples were submitted to the Integrated Genomics Operation (IGO) at MSKCC for quality and quantity analysis, library preparation and sequencing. DNA quality mas measured with Tapestation 2200. All samples had a DNA Integrity Number (DIN) >6. After PicoGreen quantification,~200 ng of genomic DNA were used for library construction using the KAPA Hyper Prep Kit (Kapa Biosystems KK8504) with eight cycles of PCR. After sample barcoding, 2.5 ng-1µg of each library were pooled and captured by hybridization with baits specific to either the HEME-PACT (Integrated Mutation Profiling of Actionable Cancer Targets related to Hematological Malignancies) assay, designed to capture all protein-coding exons and select introns of 576 (2.88Mb) commonly implicated oncogenes, tumor suppressor genes *Cheng et al., 2015* and/or HEME/BRAIN-PACT (716 genes, 3.44 Mb, *Supplementary file 2*) an expanded panel that included additional custom targets related to neurological diseases including, Alzheimer's Disease, Parkinson's Disease, Amyotrophic Lateral Sclerosis (ALS), and others (*Supplementary file 1*; *Bras et al., 2012*; *Renton et al., 2014*; *Karch et al., 2014*; *Karch and Goate, 2015*; *Turner et al., 2013*; *Ferrari et al., 2015*; *Kouri et al., 2015*; *Scholz and Bras, 2015*; *Nalls et al., 2014*). To simplify, in the manuscript, the combined panel is referred to as 'BRAIN-PACT.' In *Supplementary file 3*, 'Heme-only' or 'Brain-only' is indicated in the cases for which only one or the other panels were used. Capture pools were sequenced on the HiSeq 4000, using the HiSeq 3000/4000 SBS Kit (Illumina) for PE100 reads. Samples were sequenced to a mean depth of coverage of 1106 x (Control samples: 1071 x, AD samples 1100 x). For detailed information on the sample quality control checks used to avoid potential sample and/or barcode mix-ups and contamination from external DNA, see *Cheng et al., 2015*.

## Mutation data analysis

The data processing pipeline for detecting variants in Illumina HiSeq data is as follows. First, the FASTQ files are processed to remove any adapter sequences at the end of the reads using cutadapt (v1.6). The files are then mapped using the BWA mapper (bwa mem v0.7.12). After mapping the SAM files are sorted and read group tags are added using the PICARD tools. After sorting in coordinate order the BAM's are processed with PICARD MarkDuplicates. The marked BAM files are then processed using the GATK toolkit (v 3.2) according to best practices for tumor normal pairs. They are first realigned using ABRA (v 0.92) and then the base quality values are recalibrated with the Base-QRecalibrator. Somatic variants are then called in the processed BAMs using MuTect (v1.1.7) for SNV and ShearwaterML (*Martincorena et al., 2015*; *Martincorena and Campbell, 2015*; *Martincorena et al., 2018*). **muTect (v1.1.7):** to identify somatic variants and eliminate germline variants, we run the pipeline as follows: PU.1, DN and Blood samples against matching-NeuN samples, and NeuN samples

against matching-PU.1. In addition, we ran all samples against a Frozen-Pool of 10 random genomes. We selected Single Nucleotide Variations (SNVs) [Missense, Nonsense, Splice Site, Splice Regions] that were supported by at least four or more mutant reads and with coverage of 50 x or more. Fill-out file for each project (~27 samples per sequencing pool), were used to exclude by manual curation, variants with high background noise. This resulted in 428 variants (Missense, Nonsense, Splice_site, Splice_Region).

ShearwaterML, was used to look for low allelic frequency somatic mutations as it has been shown to efficiently call variants present in a small fraction of cells with true positives being ~90%. Briefly, the basis of this algorithm is that is uses a collection of deep-sequenced samples to learn for each site a base-specific error model, by fitting a beta-binomial distribution to each site combining the error rates across all normal samples both the mean error rate at the site and the variation across samples, and comparing the observed variant rate in the sample of interest against this background model using a likelihood-ratio test. For detailed description of this algorithm please refer to *Martincorena et al., 2015*; *Martincorena et al., 2018*. In our data set, for each cell type (NeuN, DN, PU.1) we used as 'normal' a combination of the other cell types, i.e., PU.1 vs NeuN +DN, DN vs NeuN +PU.1, NEUN vs PU.1+DN, Blood vs NeuN +DN. Since all samples were processed and sequenced using the same protocol, we expect the background error to be even across samples. More than 400 samples were used as background leading to an average background coverage >400.000 x. Resulting variants for each cell type were filtered out as germline if they were present in more than 20% of all reads across samples. Additionally, variants with coverage of less than 50 x and more than 35% variant allelic frequency (VAF) were removed from downstream analysis. p-values were corrected for multiple testing using Benjamini & Hochberg's False Discovery Rate (FDR) *Reiner et al., 2003* and a q-value of cutoff of 0.01 was used to call somatic variants. Variants were required to have a least one supporting read in each strand. Somatic variants within 10 bp of an indel were filtered out as they typically reflect mapping errors. We selected Single Nucleotide Variations (SNVs) [Intronic, Intergenic, Missense, Nonsense, Splice Site, Splice Regions] that were supported by at least 4 or more mutant reads and annotated them using VEP. Finally, to reduce the risk of SNP contamination, we excluded variants with a MAF (minor allelic frequency) cutoff of 0.01 using the gnomeAD database. This resulted in 509 SNVs.

We compared the final mutant calls from Muetct1 and ShearwaterML and found that 30% of the events (111 variants) that were called by MuTect1 were also called by ShearwaterML. Overall a total of 826 variants (*Supplementary file 3*) were found, with a mean coverage at the mutant site of 668.3 X (10% percentile: 276 X, 90% percentile: 1181 X) and a mean of 29.1 mutant reads (10% percentile: 4, 90% percentile: 52), with 84% of mutated supported by at least five mutant reads (*Supplementary file 3*). The median allelic frequency was ~1.34% (*Supplementary file 3*). Negative results for matching brain negative samples were confirmed in 100% of samples at a mean depth of ~5000 x (range 648–23.000 x) (*Supplementary file 3*), confirming nuclei sorting purity of >95% for PU.1$^+$, DN, and NEUN$^+$ populations.

## Validation of variants by droplet-digital-PCR (ddPCR)

We performed validation of ~11% of unique variants (69/760) by droplet-digital PCR (ddPCR) on pre-amplified DNA or on libraries (in the cases where DNA was not sufficient). Around 15% (15/69) of the variants analyzed by ddPCR were called by ShearwaterML, ~44% (34/69) were called by Mutect1 and 40% (24/69) by both ShearwaterML +Mutect1. Altogether we confirmed 62/69 of variants tested (~90%). In addition, 61 assays (from variants detected in PU.1+nuclei) were tested in paired cell types isolated from the same brain region. Assays were also run in matching blood when available. The mean depth of ddPCR was ~5000 x and mutant counts of three or more were considered positive. VAF obtained by ddPCR correlated with original VAF by sequencing (R2 0.93, p<0.0001). For **KRAS_G12D:** Bio-Rad validated assay (Unique Assay ID: dHsaMDV2510596) and **MTOR_Arg1616His_c.4847G>A:** Bio-Rad validated assay (Unique Assay ID: dHsaMDV2510596) were used. The remaining assays were designed and ordered through Bio-Rad. For setting-up the right conditions for newly designed assays, cycling conditions were tested to ensure optimal annealing/extension temperature as well as optimal separation of positive from empty droplets. All reactions were performed on a QX200 ddPCR system (Bio-Rad catalog # 1864001). When possible, each sample was evaluated in technical duplicates or quartets. Reactions contained 10 ng gDNA, primers and probes, and digital PCR Supermix for probes

(no dUTP). Reactions were partitioned into a median of ~31,000 droplets per well using the QX200 droplet generator. Emulsified PCRs were run on a 96-well thermal cycler using cycling conditions identified during the optimization step (95 °C 10'; 40–50 cycles of 94 °C 30' and 52–56°C 1'; 98 °C 10'; 4 °C hold). Plates were read and analyzed with the QuantaSoft software to assess the number of droplets positive for mutant DNA, wild-type DNA, both, or neither. ddPCR results are listed in *Supplementary file 3*.

## Classification of variants

To classify somatic variants according to their pathogenicity we did as follows: Variants were classified as 'pathogenic (P-SNV)' if reported as 'pathogenic/likely pathogenic' by ClinVar *Landrum et al., 2014* and/or 'oncogenic/predicted oncogenic/likely oncogenic' by OncoKb (*Chakravarty et al., 2017*; *Supplementary file 3*). These two databases report pathogenicity in cancer and other diseases, based on supporting evidence from curated literature (see corresponding citations in *Supplementary file 3*). We considered classical-MAPK-pathway genes those reported to be mutated in RASopathies: *BRAF, CBL, KRAS, MAP2K1, NF1, PTPN11, SOS1, RIT1, SHOC2, NRAS, RAF1, RASA1, HRAS, MAP2K2, SPRED1* (*Rauen, 2013*; *Tidyman and Rauen, 2016*; *Supplementary file 3*).

## Quantification of mutational load and statistics

We defined mutational load or mutational burden as the number of synonymous and non-synonymous somatic single-nucleotide-variant (SNV) per megabase of genome examined (*Zehir et al., 2017*). Overall, a total of 826 SNVs were detected resulting in 0.3 mutations/Mb sequenced. As detailed in the manuscript, the mutational load varies considerably across cell types and patients. To quantify mutational load we took into consideration the panel used for sequencing each sample: HEME-PACT (2.88 Mb) or the extended panel BRAIN-PACT (3.44 Mb) (see *Supplementary file 2*). Therefore, the number of mutations was normalized by the number of Mb sequenced for that specific sample. In the cases where we calculated mutational load per patient, we averaged the mutational load of each sample from that patient for a given cell type (i.e. if for one patient, 2 PU.1 samples were sequenced, one from hippocampus and one from superior parietal cortex (with BRAIN-PACT) then the mutational load for PU.1 for that patient is the mean of the mutational load of the 2 PU.1 samples analyzed). For the quantification of 'pathogenic' variants, the same analysis is performed, quantifying only variants that are reported as pathogenic by ClinVar and/or OncoKb. Statistical significance was analyzed with GraphPad Prism (v9) and R (3.6.3). Non-parametric tests were used when data did not follow a normal distribution (Normality test: D'Agostino-Pearson and Shapiro-Wilk test). For normally distributed data, unpaired t-test was used to compare two groups and one-way, nested one-way or two-way analyses of variance (ANOVA) were used for comparing more than two groups, as indicated in the Figure legends. For data that did not have a normal distribution, the tests performed were unpaired two-tailed Mann-Whitney U test and Kruskal–Wallis test and Dunn's test for multiple comparisons. Pearson and Spearman were used for correlation analysis. In *Figure 2G*, we used multivariate logistic regression analysis to test if there was an association between Alzheimer's disease and the presence of pathogenic variants in PU.1$^+$ nuclei. We used Alzheimer's disease as a dependent variable, and age, sex, and the presence of pathogenic variant/s (Yes/No) as co-variates. In all the statistical tests, significance was considered at $p < 0.05$. For Venn Diagram plots, we used (*Bardou et al., 2014*).

## Mixed-effects modeling of somatic P-SNV burden

To evaluate the correlation between P-SNV burden and disease status (non-dementia controls and AD) after adjusting for other factors such as individual donor and age, we performed linear mixed-effects regression modeling using the nlme package in R. In the subsequent analysis, we also tested another framework implemented in the lme4 package and confirmed similar findings. We estimated the linear mixed model via maximum likelihood method with nlminb optimizer. The model included individual donor as a random effect. We also tested if inclusion of other co-variates (i.e. sex, anatomical location of the brain, and source biobank of brain samples) as random effects inproved the overall model fitting via likelihood ratio test. However, none of these co-variates improved the model fitting ($p > 0.99$). Thus, we used a relatively simple model that incorporated disease status and age as fixed effects and donor as random effect. The total explanatory power of the final model is substantial (conditional $R^2 = 0.48$). To assess the significance of age or disease status in predicting P-SNV burden,

we constructed another model that does not incorporate the variable as fixed effect, and compared the two models via likelihood ratio test. The variable was considered significantly associated with P-SNV burden when the p-value was below 0.05 and the AIC increased after removing the variable.

Pathway enrichment analysis of genes target of variants was performed using Metascape *Zhou et al., 2019* and the following ontology sources: KEGG Pathway (*Huang et al., 2009a*; *Huang et al., 2009b*), GO Molecular function (*Ashburner et al., 2000*; *The Gene Ontology, 2019*), Reactome Gene Sets *Ashburner et al., 2000*; *The Gene Ontology, 2019*, and Canonical Pathways (*Schaefer et al., 2009*). The list of 716 genes from the targeted panel were used as the enrichment background. Terms with a p-value <0.05, a minimum count of 3, and an enrichment factor >1.5 (the enrichment factor is the ratio between the observed counts and the counts expected by chance) are shown. p-values are calculated based on the cumulative hypergeometric distribution *Zar, 2010*.

## Expression of target genes in microglia

To evaluate the expression levels of the genes identified in this study as target of somatic variants, we consulted a publicly available database (https://www.proteinatlas.org/), and also plotted their expression as determined by RNAseq in two studies (Galatro et al. GSE99074, *Galatro et al., 2017*, and *Gosselin et al., 2017*) (Supplementary Fig S3 and Figure S2). For data from Galatro et al. (GSE99074, *Galatro et al., 2017*), normalized gene expression data and associated clinical information of isolated human microglia (N=39) and whole brain (N=16) from healthy controls were downloaded from GEO. For data from *Gosselin et al., 2017*, raw gene expression data and associated clinical information of isolated microglia (N=3) and whole brain (N=1) from healthy controls were extracted from the original dataset. Raw counts were normalized using the DESeq2 package in R (*Love et al., 2014*).

## Nuclei isolation from frozen brain samples for snRNA-seq

For snRNA-seq studies, we only selected samples with a RIN score in whole tissue of six or more. All samples were handled and processed under Air Clean PCR Workstation. About 250–400 mg of frozen brain tissues were homogenized with a sterile Dounce tissue grinder using a sterile homogenization buffer to isolate cell nuclei (250 mM Sucrose, 25 mM KCL, 5 mM MgCl2, 10 mM Tris buffer pH 8.0, 0.1% (v/v) Triton X-100, 3 µM DAPI, Nuclease Free Water and 20 U/ml of Superase-In RNase inhibitor, and 40 U/ml RNasin ribonuclease inhibitor). Homogenate was filtered in a 40 µm cell strainer and centrifuged 800 g 8 min 4 °C. To clean-up the homogenate, we performed a iodixanol density gradient centrifugation as follows: pellet was gently mixed 1:1 with iodixanol medium at 50% (50% Iodixanol, 250 mM Sucrose, 150 mM KCL, 30 mM MgCl2, 60 mM Tris buffer pH 8.0, Nuclease Free Water) and homogenization buffer. This solution layered to a new tube containing equal volume of iodixanol medium at 29% and centrifuged 13.500 g for 20 min at 4 °C. Nuclei pellet was resuspended in FACS buffer with RNAse inhibitors (0.5% BSA, 2 mM EDTA, Superase-In RNase inhibitor and 40 U/ml RNasin ribonuclease inhibitor) and centrifuged 800 g 5 min, 4 °C. Nuclei pellet was fixed with 90% ice-cold methanol and incubated for 10 min on ice, followed by a centrifugation at 1300 g (without brakes, which improves with nuclei recovery after fixation). The pellet was resuspended in permeabilization buffer (6% BSA, Superase-In RNase inhibitor 20 U/mL, RNasin ribonuclease inhibitor 40 U/mL and 0.05% Triton) followed by a centrifugation at 1300 g. Sample was incubated with anti-Pu.1 antibody (microglia marker 1:50, Pu.1-AlexaFluor 647, 9G7 Cell Signeling) in permeabilization buffer. After a wash with FACS buffer sample were ready for sorting. Nuclei are FACS-sorted in a BD FACS Aria with a 100 µm nozzle and a sheath pressure 20 psi, operating at ~1000 events per second. Nuclei were sorted into 1.5 ml certified RNAse, DNAse DNA, ATP, and Endotoxins tubes containing 100 µl of sterile PBS. For each population, we sorted >$10^5$ nuclei into FACS buffer.

## SnRNA-seq library preparation and sequencing

The single-nuclei RNA-seq of FACS-sorted nuclei suspensions was performed on Chromium instrument (10 X genomics) following the user guide manual (Reagent Kit 3' v3.1). Each sample, containing approximately 10,000 nuclei at a final dilution of ~1000 cells/µl was loaded onto the cartridge following the manual. The individual transcriptomes of encapsulated cells were barcoded during RT step and resulting cDNA purified with DynaBeads followed by amplification per manual guidelines. Next, PCR-amplified product was fragmented, A-tailed, purified with 1.2 X SPRI beads, ligated to the sequencing adapters, and indexed by PCR. The indexed DNA libraries were double-size purified (0.6–0.8 X) with

SPRI beads and sequenced on Illumina NovaSeq S4 platform (R1 – 26 cycles, i7 – 8 cycles, R2 – 70 cycles or higher). Sequencing depth was ~200 million reads per sample on average. FASQ files were processed using SEQC pipeline *Azizi et al., 2018* for quality control, mapping to GRCH38 reference genome, and log2 transformation of the data with the default SEQC parameters to obtain the gene-cell count matrix.

## SnRNA-seq analysis

Seurat v4.0.3 with default parameters was used to perform sctransform (SCT) normalization, integration and Uniform Manifold Approximation and Projection (UMAP) for dimensionality reduction. The FindClusters function was used for cell clustering. To improve clustering, all samples were analyzed in an integrated analysis, based on canonical correlation analysis (CCA). Cell types were annotated using the top 500 DEGs of each cell type in a human cortex database. Data can be accessed at https://weillcornellmed.shinyapps.io/Human_brain/. The removal of doublets using DoubletFinder and cells with high mitochondrial content (>10% mitochondrial RNA) yielded between 6437 and 9241 nuclei per patient and sample. Microglia represented 94 ± 3% of total cells. Unique Molecular Identifiers (UMIs) per nucleus and gene count per nucleus were comparable between donors. Integrated_snn at resolution 0.2 outlined 16 microglia clusters. Except for cluster 13 consisting of 97% of cells from the healthy control Control 11_AG, all donors and samples were represented in every cluster. One cluster contained few cells (0.84% of total microglia, for an average of 6.20 ± 1.60% for other clusters) and was marked by a low number of cluster-enriched genes and was excluded from further analyses. For pathway enrichment analysis, genes were pre-ranked using differential expression analysis in SCANPY *Wolf et al., 2018* with Wilcoxon rank-sum method. Statistical analysis were performed using the fgsea-Multilevel function in fgsea R package *Korotkevich et al., 2019* for HALLMARK and KEGG pathways. Gene sets with p-value <0.05 and adjusted p-value <0.25 were selected and visualized using ggpubr and ggplot2 *Hadley, 2016* R package. For the variant analysis of AD52_HIP harboring a KRAS[A59G] (c.176C>G) clone, Integrative Genomics Viewer (IGV) software was used to display sequencing reads at KRAS c.176C (exon 3; GRCh38 chr12:25,227,348). Cells within each cluster were identified based on the 16-digit barcodes from SEQC-aligned reads. Barcodes were converted to the 10 X Genomics format and used to sample reads from each cluster within the original BAM file. BAM subsets for each cluster were read with IGV and reads with identical UMIs were filtered out to account for amplification bias.

## Whole-exome-sequencing and analysis

Remaining libraries from a selected group of PU.1 and NEUN samples sequenced with BRAIN-PACT (see above) were sequenced by Whole-Exome-Sequencing (WES). Matching NEUN samples were sequenced to extract the germline variants. Around 100 ng of library were captured by hybridization using the xGen Exome Research Panel v2.0 (IDT) according to the manufacturer's protocol. PCR amplification of the post-capture libraries was carried out for 12 cycles. Samples were run on a NovaSeq 6000 in a PE100 run, using the NovaSeq 6000 S4 Reagent Kit (200 Cycles) (Illumina). Samples were covered to an average of 419 X. The data processing pipeline for detecting variants in Novaseq data is as follows. First the FASTQ files are processed to remove any adapter sequences at the end of the reads using cutadapt (v1.6). The files are then mapped using the BWA mapper (bwa mem v0.7.12). After mapping the SAM files are sorted and read group tags are added using the PICARD tools. After sorting in coordinate order the BAM's are processed with PICARD MarkDuplicates. The marked BAM files are then processed using the GATK toolkit (v 3.2) according to the best practices for tumor normal pairs. They are first realigned using ABRA (v 0.92) and then the base quality values are recalibrated with the BaseQRecalibrator. Somatic variants are then called in the processed BAMs using muTect (v1.1.7) for SNV and the Haplotype caller from GATK with a custom post-processing script to call somatic indels. The full pipeline is available here https://github.com/soccin/BIC-variants_pipeline (copy archived at *Socci, 2022*) and the post processing code is at https://github.com/soccin/Variant-PostProcess, (copy archived at *Socci, 2025*). We selected Single Nucleotide Variants (SNVs) [Missense, Nonsense, Splice Site, Splice Regions] that were supported by at least 8 or more mutant reads, variant allelic frequency above 5% and with coverage of 50 x. Annotation was performed using VEP. Finally, to reduce the risk of SNP contamination, we excluded variants with a MAF (minor allelic frequency) cutoff of 0.01 using the genomeAD database. Variants were classified as 'candidate pathogenic' when

SNV is predicted to affect the protein as determined by PolyPhen-2 (possibly and probably damaging) and SIFT (deleterious) and CADD-MSC (high) and FATHMM-XF Functional Analysis through Hidden Markov Models (pathogenic) (*Rogers et al., 2018*; *Supplementary file 5*).

## Cell lines

### HEK293T cell culture and transfection

HEK 293T cells (ATCC) were maintained in Dulbecco's modified Eagle's medium (Mediatech, Inc) supplemented with 10% fetal bovine serum (Sigma) and 1000 IU/ml penicillin, 1000 IU/ml streptomycin.

### BV2 microglial cell line

BV2 murine microglial cells were cultured in Dulbecco's modified Eagle's medium (DMEM) High Glucose medium (Gibco), Glutamax (Gibco), sodium pyruvate, 1% non-essential amino acids (Invitrogen), and 10% heat-inactivated fetal bovine serum (FBS, EMD Millipore). For MAPK activation experiments, cells were treated with M-CSF1 100 ng/ml for 5 min.

### MAC cell lines

Mouse primary CSF-1 dependent macrophages immortalized with the SV-U19-5 retrovirus *Xiong et al., 2011* were a gift of Dr. E. R. Stanley (Albert Einstein College of Medicine, Bronx, NY). They were cultured in RPMI 1640 medium with Glutamax (Gibco), 10% heat-inactivated fetal bovine serum (FBS, EMD Millipore) and 100 ng/mL recombinant CSF-1 (gift from Dr. E. R. Stanley). Growth medium was renewed every second day. When confluency reached 80%, cells were passaged by cell scraping and plated at $5 \times 10^4$ cells/cm$^2$ in tissue culture treated plates. For signaling pathway analyses, cell proliferation assays or collection for RNA sequencing, cells were plated 1 d prior at $5 \times 10^4$ cells/cm$^2$ in medium containing 10 ng/mL CSF-1 for lines expressing wild-type (WT) and mutant CBL, RIT1, and KRAS proteins and 100 ng/mL CSF-1 for lines expressing WT and mutant PTPN11 proteins. Cells were grown at 37 °C and 5% CO2.

### Human-induced pluripotent stem cell (hiPSC) culture

hiPSC lines were derived from peripheral blood mononuclear cells (PBMCs) of a healthy donor. Written informed consent was obtained according to the Helsinki convention. The study was approved by the Institutional Review Board of St Thomas' Hospital; Guy's hospital; the King's College London University; the Memorial Sloan Kettering Cancer Center and by the Tri- institutional (MSKCC, Weill-Cornell, Rockefeller University) Embryonic Stem Cell Research Oversight (ESCRO) Committee. hiPSC were derived using Sendai viral vectors (Thermo Fisher Scientific; A16517). Newly derived hiPSC clones were maintained in culture for 10 passages (2–3 mo) to remove any traces of Sendai viral particles. Over 90% of hiPSCs in the derived lines expressed high levels of the pluripotency markers NANOG and OCT4 by flow cytometry. The C12 hiPSC WT line was engineered to carry a CBL p.C404Y, c.1211G>A heterozygous variant at the endogenous CBL locus. HiPSCs of passage 25–35 were cultured on confluent irradiated CF1 mouse embryo fibroblasts (MEFs, Gibco) in hiPSC medium consisting of knock-out DMEM (Invitrogen), 10% knock-out-Serum Replacement (Invitrogen), 2 mM L-glutamine (Gibco), 100 U/mL penicillin-streptomycin (Invitrogen), 1% non-essential amino acids (Invitrogen), 0.1 mM β-mercaptoethanol (R&D). hiPSC medium was supplement with 10 ng/mL bFGF (PeproTech) and changed every second day. Two days before culture with hiPSCs, MEFs were plated at 20,000 cells/cm$^2$ in DMEM supplemented with 10% heat-inactivated fetal bovine serum (FBS, EMD Millipore), 100 U/mL penicillin- streptomycin (Invitrogen), 1% non-essential amino acids (Invitrogen) and 0.1 mM β-mercaptoethanol (R&D Systems) on 150 mm tissue culture plates coated with 0.1% gelatin (Sigma). hiPSCs were passaged weekly with 250 U/mL collagenase type IV (Thermo Fisher Scientific) at a 1:4 to 1:6 ratio onto MEF cells in hiPSC medium supplemented with 10 μM Rock inhibitor (Y-27632 dihydrochloride, Sigma). Cells were maintained at 37 °C and 5% CO2 and they were routinely tested for mycoplasma and periodically assessed for genomic integrity by karyotyping. Microglia-like cells were obtained from hiPSCs using an embryoid body (EB)-based protocol as previously described *Lachmann et al., 2015*. Briefly, hiPSC were loosened with 250 U/mL collagenase type IV (Thermo Fisher Scientific) and lifted with cell scraping. For EBs formation, hiPSC colonies were transferred to suspension plates on an orbital shaker in hiPSC medium supplemented with 10 μM Rock

inhibitor (Y-27632 dihydrochloride, Sigma). After 6 d, EBs were transferred to 6-wells tissue culture treated plates in STEMdiff APEL 2 medium (Stem Cell Technology) with 5% Protein Free Hybridoma Media (Gibco), 100 U/mL penicillin-streptomycin (Invitrogen), 25 ng/mL IL-3 (Peprotech) and 50 ng/mL CSF-1 (Peprotech). Microglia-like cells were harvested every week from the supernatant of EBs cultures. Collected microglia-like cells were used immediately for signaling pathway analyses or plated for 6–7 d in RPMI 1640 medium with Glutamax supplement (Gibco), 10% heat-inactivated fetal bovine serum (FBS, EMD Millipore) and 100 ng/mL human recombinant CSF-1 (Peprotech) in tissue culture plates for cytology, flow cytometry, RNA sequencing, and supernatant analyses of cytokines release. Microglia like cells differentiation was monitored by May-Grunwald Giemsa staining and flow cytometry analyses of myeloid markers.

All cells used in the study were routinely tested for mycoplasma and were negative.

## Plasmids used in in-vitro studies (HEK293, BV2, and MAC lines)

The expression vectors for Flag-tagged CHK2 kinase and RIT1 were from Sino Biological and Origene, respectively. The vector encoding pcDNA3-HA-tagged c-Cbl was a kind gift from Dr. Nicholas Carpino (Stony Brook). RIT1$^{M90I,}$ RIT1$^{F82L}$, CBL$^{I383M}$, CBL$^{C404Y}$, CBL$^{C416S}$, CBL$^{C384Y}$, CBL$^{Y371H}$ were generated by site-directed mutagenesis using the QuikChange Kit (Agilent). pHAGE_puro was a gift from Christopher Vakoc (Addgene plasmid # 118692; http://n2t.net/addgene:118692; RRID:Addgene_118692) (*Lu et al., 2018*). pHAGE-KRAS was a gift from Gordon Mills & Kenneth Scott (Addgene plasmid # 116755; http://n2t.net/addgene:116755; RRID:Addgene_116755) (*Ng et al., 2018*) *Cheng et al., 2015*. pHAGE-PTPN11 was a gift from Gordon Mills & Kenneth Scott (Addgene plasmid # 116782; http://n2t.net/addgene:116782; RRID:Addgene_116782) (*Ng et al., 2018*). pHAGE-PTPN11-T73I was a gift from Gordon Mills & Kenneth Scott (Addgene plasmid # 116647; http://n2t.net/addgene:116647; RRID:Addgene_116647) (*Ng et al., 2018*). pDONR223_KRAS_p.A59G was a gift from Jesse Boehm & William Hahn & David Root (Addgene plasmid # 81662; http://n2t.net/addgene:81662; RRID:Addgene_81662) (*Kim et al., 2016*), Phage-CBL, Phage-CBL$^{I383M}$, Phage-CBL$^{C404Y}$, Phage-CBL$^{C416S}$, Phage-RIT1, Phage-RIT1$^{M90I}$ and Phage-RIT1$^{F82L}$ and Phage-KRAS$^{A59G}$ were generated by Azenta Life Sciences via a PCR cloning approach. pHAGE-CBL$^{C384Y}$ plasmid was generated at Azenta Life Science by targeted mutagenesis of pHAGE-CBL.

## Generation of mutant lines

HEK293 were transfected 24 hr after plating with 2.5 µL of Mirus Transit LT1 per µg of DNA. Cells were harvested and lysed 48 hr after transfection using a buffer containing 25 mM Tris, pH 7.5, 1 mM EDTA, 100 mM NaCl, 1% NP-40, 10 µg/ml leupeptin, 10 µg/ml aprotinin, 200 µM PMSF, and 0.2 mM Na$_3$VO$_4$. For EGF stimulation, the media was replaced 24 hr after transfection with DMEM containing 1% FBS and antibiotics. After a further 24 hr in this starvation media, the cells were stimulated with 50 ng/ml EGF for 5 min at 37 °C.

## Lentiviral production and transduction of BV2 and MAC cell lines

For BV2 cell line, cells were transduced for 24 hr without the presence of Vpx VLPs and selected with 2.5 µg/mL puromycin (Fisher Scientific). For 'MAC' lines, Vpx-containing virus-like particles (Vpx VLPs) were produced by transfection of HEK293T cells with 4.8 µg VSV-g plasmid and 31.2 µg pSIV3/Vpx plasmid, a gift from Dr. M. Menager (Imagine Institute, Paris, France) using TransIT-293 Transfection Reagent (Mirus Bio, Fisher Scientific). Forty-eight hours after transfection, the supernatant containing Vpx VLPs was collected and used immediately for lentiviral transduction of macrophages. Viral supernatants were obtained by transfection of HEK293T cells using X-tremeGENE HP DNA Transfection Reagent (Sigma). Packaging vectors used were psPAX2 (gift from Didier Trono Addgene plasmid # 12260; http://n2t.net/addgene:12260; RRID:Addgene_12260) and pMD2.G (gift from Didier Trono, Addgene plasmid # 12259; http://n2t.net/addgene:12259; RRID:Addgene_12259). Cells were transduced for 24 hr in presence of Vpx VLPs. Transduced macrophages were selected with 5 µg/mL puromycin (Fisher Scientific).

## Generation of the CBL$^{+/C404Y}$ and isogenic WT hiPSC lines

The CBL$^{C404Y}$ (c.1211 G>A) variant was inserted at the endogenous locus in the C12 WT hiPSC using Cytidine base editing (CBE) with CBE enzyme BE3-FNLS (*Zafra et al., 2018*). Briefly, the sgRNA for CBE was designed to target the non-coding strand and introduce the position 6 'C-to-T' conversion, to create the G-to-A conversion on the coding strand. The sgRNA target sequence was cloned into the pSPgRNA (Addgene plasmid # 47108) *Perez-Pinera et al., 2013* to make the gene targeting construct. To introduce the CBL C404Y variants, the WT hiPSC (C12) were dissociated using Accutase (Innovative Cell Technologies) and electroporated (1×10⁶ cells per reaction) with 4 µg sgRNA-construct plasmid and 4 µg CBE enzyme coding vector BE3-FNLS (Addgene plasmid # 112671) (*Zafra et al., 2018*) using Lonza 4D-Nucleofector and the Nucleofector solution (Lonza V4XP-3034) following our previously reported protocol (*Zhong et al., 2020*). The cells were then seeded, and 4 d later, the hiPSC were dissociated into single cells by Accutase and re-plated at a low density (four per well in 96-well plates) to get the single-cell clones. Ten days later, individual colonies were picked, expanded and analyzed by PCR and DNA sequencing to identify the clones carried the desired CBL$^{C404Y}$ heterozygous variant and the isogenic WT control clones. The sgRNA target, PCR, and sequencing primers are listed below.

| | sgRNA target | PCR-Forward primer | PCR-Reverse primer (used for sequencing) |
|---|---|---|---|
| CBL$^{C404Y}$ | TAAGACAGGATGTGCACATG | TGGGCTCCACATTCCAACTA | GCCCTGACCTTCTGATTCCT |

## Western blotting

For HEK293 cells, lysates were resolved by SDS-PAGE, transferred to PVDF membranes, and probed with the appropriate antibodies. Horseradish peroxidase-conjugated secondary antibodies (GE Healthcare) and Western blotting substrate (Thermo) were used for detection. For anti-Cdc42 immunoprecipitation experiments, cell lysates (1 mg total protein) were incubated overnight with 1 µg of anti-Cdc42 antibody (Santa Cruz) and 25 µL of protein A agarose (Roche) at 4°C. Anti-Flag immunoprecipitations were done with anti-Flag M2 affinity resin (Sigma). The beads were washed three times with lysis buffer, then eluted with SDS-PAGE buffer, and resolved by SDS-PAGE. The proteins were transferred to PVDF membrane for Western blot analysis. Antibodies used are Phospho-p44/42 MAPK (pErk 1/2) (Thr202/Tyr204) is from Cell Signaling #4370, total p44/42 MAPK (Erk1/2) is from Cell Signaling #9102, HA tag from Millipore # 05–904, Flag antibody is from Sigma (#A8592), pCHEK2 (T383) antibody is from Abcam, #ab59408, Cdc42 antibody is from Santa Cruz (#sc87) and Anti-γ-Tubulin antibody (Sigma T6557). **For Immunoprecipitation Kinase assay in HEK293T cells,** cell lysates (1 mg protein) were incubated overnight with 30 µL of anti-Flag M2 affinity resin on a rotator at 4°C, then washed three times with Tris-buffered saline (TBS). A portion of each sample was eluted with SDS-PAGE sample buffer and analyzed by anti-Flag Western blotting. The remaining sample was used for a radioactive kinase assay. The immunoprecipitated proteins were incubated with 25 µL of reaction buffer (30 mM Tris, pH 7.5, 20 mM MgCl$_2$, 1 mg/mL BSA, 400 µM ATP), 650 µM CHKtide peptide (KKKVRSGLYRSPSMPENLNRPR, SignalChem), and 50–100 cpm/pmol of [γ *Zar, 2010*-P] ATP at 30 °C for 15 min. The reactions were quenched using 45 µL of 10% trichloroacetic acid. The samples were centrifuged and 30 µL of the reaction was spotted onto Whatman P81 cellulose phosphate paper. After washing with 0.5% phosphoric acid, incorporation of radioactive phosphate into the peptide was measured by scintillation counting. **For MAC lines and hiPSC-derived cells,** cell lysates obtained with RIPA buffer +1:1000 Halt Protease and Phosphatase Inhibitor Cocktail (Thermo Fisher Scientific) were sonicated three times for 30 s at 4 °C (Bioruptor, Diagenode). Protein quantification of supernatant was done with Precision Red Advanced Protein Assay (Cytoskeleton). Proteins were boiled for 5 min at 95 °C in NuPAGE LDS sample buffer (Invitrogen) and separated in NuPAGE 4–12% Bis-Tris Protein Gel (Invitrogen) in NuPAGE MES SDS Running Buffer (Invitrogen). Electrophoretic transfer to a nitrocellulose membrane (Thermo Fisher Scientific) was done in NuPAGE Transfer Buffer (Invitrogen). Blocking was performed for 60 min in TBS-T +5% nonfat milk (Cell Signaling) and incubated with primary antibodies at 4°C: rabbit anti-p44/42 MAPK (ERK1/2) (Cell Signaling; 1:1000); rabbit anti-P-p44/42 MAPK (Cell Signaling, 1:1000); rabbit anti-c-CBL (Cell signaling, 1:1000); rabbit anti-RIT1 (Abcam, 1:1000); mouse anti-KRAS (clone 3B10-2F2, Sigma, 1 µg/mL); mouse anti-Actin (clone MAB1501, Sigma, 1:10,000). Primary antibodies were detected using the secondary anti-rabbit

IgG HRP-linked (Cell Signaling, 1:1000) or the anti-mouse IgG HRP-linked (Cell Signaling, 1:1000) were used to detect primary antibodies, with SuperSignal West Femto Chemiluminescent Substrate (Thermo Fisher Scientific) using a ChemiDoc MP Imaging System (Bio-Rad). pERK/ERK ratios were measured with ImageJ software.

## DNA/RNA isolation, dd-PCR, and RTqPCR in MAC lines and hiPSC-derived cells

Genomic DNA was extracted using QIAamp DNA Micro Kit (50) (Qiagen), following the manufacturer's instructions. Total RNA was extracted using RNeasy Mini kit (Qiagen), following the manufacturer's instructions. cDNA was generated by reverse transcription using Invitrogen SuperScript IV Reverse Transcriptase (Invitrogen) with oligo(dT) primers. The TaqMan gene expression assays used were c-CBL FAM (Hs01011446_m1), CBLb FAM (Hs00180288_m1) and GAPDH VIC (Hs02786624_g1) (Thermo Fisher Scientific). RT-qPCR was performed using Applied Biosystems TaqMan Fast Advanced Master Mix (Thermo Fisher Scientific) and a QuantStudio 6 Flex Real-Time PCR System (Thermo Fisher Scientific). The results were normalized to GAPDH. For droplet PCR analyses, assays specific for the detection of I383M, C384Y, C404Y and C416S in CBL and F82L and M90I in RIT1, A59G in KRAS, and corresponding WT sequences (listed below) were obtained from Bio-Rad. Cycling conditions were tested to ensure optimal annealing/extension temperature as well as optimal separation of positive from empty droplets. Optimization was done with a known positive control. After PicoGreen quantification, 2.6–9 ng gDNA or cDNA were combined with locus-specific primers, FAM- and HEX-labeled probes, HaeIII, and digital PCR Supermix for probes (no dUTP). All reactions were performed on a QX200 ddPCR system (Bio-Rad catalog # 1864001) and each sample was evaluated in technical duplicates. Reactions were partitioned into a median of ~19,000 droplets per well using the QX200 droplet generator. Emulsified PCRs were run on a 96-well thermal cycler using cycling conditions identified during the optimization step (95 °C 10'; 40 cycles of 94 °C 30' and 52–55°C 1'; 98 °C 10'; 4 °C hold). Plates were read and analyzed with the QuantaSoft software to assess the number of droplets positive for mutant or wild-type DNA.

| Assay name | Assay ID |
| --- | --- |
| CBL_I383M | dHsaMDS675699482 |
| CBL_C384Y | dHsaMDS386449640 |
| CBL_C404Y | dHsaMDS437459772 |
| CBL_mRNA_C404Y | dMDS334857054 |
| CBL_C416S | dHsaMDS613275900 |
| RIT1_ F82L | dMDS959028273 |
| RIT1_M90I | dHsaMDS133045056 |
| KRAS_A59G | dHsaMDS581417660 |

Flow cytometry analyses for surface antigens CSF1-R, CD11b, MRC1, α5β3, CD11c, Tim4, HLA-DR, CD45, CD14, NGFR, EGFR, CD36, and SIRPα were performed using PE-conjugated anti-CD115 (CSF1-R) (clone 9-4D2, BD Pharmingen), PE/Cy7-conjugated anti-CD11b (clone ICRF44, Biolegend), Alexa Fluor 488-conjugated anti-CD206 (MRC1) (clone 19.2, Thermo Fisher Scientific), PE-conjugated anti-integrin α5β3 (clone 23C6, R&D systems), PE/Cy5-conjugated anti-CD11c (Clone B-ly6, BD Pharmigen), APC-conjugated anti-Tim4 (Clone 9F4, BioLegend), PE/Cy7-conjugated anti-HLA-DR (clone G46-6, BD Pharmigen), BV650-conjugated anti-CD45 (clone HI30, BD Horizon), APC/Cy7-conjugated anti-CD14 (clone M5E2, Biolegend), PE-conjugated anti-NGFR (clone ME20.4, eBioscience), Alexa Fluor 647-conjugated anti-EGFR (clone EGFR.1, BD Pharmigen), APC/Cy7-conjugated anti-CD36 (clone 5-271, BioLegend), and APC-conjugated anti-CD172a (SIRPα) (Clone: 15 414, Thermo Fisher Scientific) antibodies. Iba1 expression was detected following fixation and permeabilization of macrophages using BD Cytofix/Cytoperm solution (BD Pharmingen). Cells were marked with Zombie Violet Viability (Biolegend). After incubation with FcR Blocking Reagent (Miltenyi Biotec), cells were stained with Alexa Fluor 555-conjugated anti-Iba1 antibody (clone E4O4W, Cell Signaling). Flow cytometry was performed using a BD Biosciences

LSR Fortessa flow cytometer with Diva software. Data were analyzed using FlowJo (BD Biosciences LLC).

## Cell proliferation analyses

For hiPSC-derived cells, cell suspension was filtered through a 100 μm nylon mesh (Corning) and marked with Zombie Violet Viability (Biolegend). After incubation with FcR Blocking Reagent (Miltenyi Biotec), surface receptors were labeled with PE/Cy7-conjugated anti-CD11b (clone ICRF44, Biolegend), Alexa Fluor 488-conjugated anti-CD206 (MRC1) (clone 19.2, Thermo Fisher Scientific), BV650-conjugated anti-CD45 (clone HI30, BD Horizon), APC/Cy7-conjugated anti-CD14 (clone M5E2, Biolegend) prior to EdU detection. For proliferation studies in the mouse macrophage cell lines, macrophages were incubated with 10 μM EdU (Thermo Fisher Scientific) for 2 hr at 37 °C and collected by cell scraping and marked with Zombie Violet Viability (Biolegend) prior to EdU detection. EdU detection was performed using the Click-iT Plus EdU Alexa Fluor 647 Flow Cytometry Assay Kit (Thermo Fisher Scientific), following the manufacturer's instructions. hiPSC-derived macrophages were analyzed using a BD Biosciences Aria III cell sorter and macrophages were identified as CD11b$^+$CD45$^+$CD14$^+$MRC1$^+$. The macrophage cell lines were analyzed using a BD Biosciences LSR Fortessa flow cytometer. Data were analyzed using FlowJo 10.6 (BD Biosciences LLC).

## Enzyme-linked immunosorbent assay

Supernatants of iPSC-derived microglia-like cells were analyzed for human inflammatory cytokines IL-6, TNFα, IL-1β, IFNγ and for the complement C3 and complement Factor H by Enzyme-linked immunosorbent assay (ELISA) at Eve Technologies (Calgary, AB).

## Bulk RNA sequencing (RNAseq)

Three biological replicates were processed for each condition/cell line. In view of RNA sequencing, phase separation in cells lysed in 1 mL TRIzol Reagent (Thermo Fisher Scientific) was induced with 200 μL chloroform and RNA was extracted from the aqueous phase using the miRNeasy Mini Kit (Qiagen) on the QIAcube Connect (Qiagen) according to the manufacturer's protocol with 350 μL input, or using the MagMAX mirVana Total RNA Isolation Kit (Thermo Fisher Scientific) on the King-Fisher Flex Magnetic Particle Processor (Thermo Fisher Scientific) according to the manufacturer's protocol with 350 μL input. Samples were eluted in 30 μL RNase-free water. After RiboGreen quantification and quality control by Agilent BioAnalyzer, 231–500 ng of total RNA with RIN values of 9.4–10 underwent polyA selection and TruSeq library preparation according to instructions provided by Illumina (TruSeq Stranded mRNA LT Kit, Illumina), with 8 cycles of PCR. Samples were barcoded and run on a NovaSeq 6000 in a PE100 run, using the NovaSeq 6000 S4 Reagent Kit (200 Cycles) (Illumina). An average of 90 million paired reads was generated per sample. Ribosomal reads represented 0–1.6% of the total reads generated and the percent of mRNA bases averaged 79%.

## Bulk RNAseq analysis

FastQ files of 2×100 bp paired-end reads were quality checked using FastQC (https://www.bioinformatics.babraham.ac.uk/projects/fastqc/, 2012). Samples with high quality reads (Phred score ≥ 30) were aligned to the *Mus musculus* genome (GRCm38.80) for the MAC lines or *Homo sapiens* (assembly GRCh38.p14) for the IPSCs lines using STAR aligner. For the MAC lines, we computed the expression count matrix from the mapped reads using HTSeq (www-huber.embl.de/users/anders/HTSeq) and one of several possible gene model databases. The raw count matrix generated by HTSeq are then be processed using the R/Bioconductor package DESeq (www-huber.embl.de/users/anders/DESeq) which is used to both normalize the full dataset and analyze differential expression between sample groups. For the ISPCs line dataset, gene quantification was performed using feature counts from the Subread package in R. Gene expression levels were normalized and log2 transformed using the Trimmed Mean of M-values (TMM) method and differential expression analysis was performed using the edgeR package in R. For hiPSC derived cells, GSEA (Hallmark, KEGG, GO, REACTOME) were performed using the fgsea package in R on a pre-ranked list (formula: sign(ogFC) * -log10(PValue)) on all expressed genes in the dataset. For the MAC cell lines dataset, GSEA was performed using gsea4.3.2 for KEGG and HALLMARK canonical pathways in MSigDB v 7.5.1. Significant genesets were selected based on an FDR ≤ 0.25.

For lists of differentially expressed genes, genes were selected with controlled False Positive Rate (B&H method) at 5% (FDR ≤ 0.05). Genes were considered upregulated/downregulated for log2 fold change >1.5 or <−1.5.

## Statistical analysis

Statistical methods are detailed in the corresponding sections above (Quantification of mutational load and statistics, Bulk RNAseq analysis, SnRNA-seq analysis) and in the Fig. legends. p-values of 0.05 and adj. p-values (FDR) of 0.25 are considered significant unless otherwise specified.

## Code availability

All code used in this study has been previously published as referenced in the method section above.

## Acknowledgements

This study was supported by grants from NIH: P30 CA008748 MSKCC core grant, 1R01NS115715-01, 1 R01 HL138090-01, and 1 R01 AI130345-01 to FG, and Basic and Translational Immunology Grants from Ludwig Center for Cancer Immunotherapy and from Cycle for Survival to FG. RV was supported by the 2018 AACR-Bristol-Myers Squibb Fellowship for Young Investigators in Translational Immuno-oncology, Grant Number 18-40-15-VICA. LW was supported by NYSTEM training award C32559GG and a Charles H Revson fellowship. Sequencing costs and analysis were covered in part by a SRA between Third Rock venture and MSKCC. The funders had no role in study design, data collection and analysis, decision to publish, or preparation of the manuscript. MAC mouse cell lines were kindly provided by Dr Richard E Stanley. Code for Shearwater ML and for single cell mRNA genotyping were provided by Dr Inigo Martincorena by Dr Noor Sohail, respectively.

## Additional information

### Funding

| Funder | Grant reference number | Author |
| --- | --- | --- |
| National Institutes of Health | P30 CA008748 | |
| NIH Office of the Director | 1R01NS115715-01 | Frédéric Geissmann |
| NIH Office of the Director | 1 R01 HL138090 01 | Frédéric Geissmann |
| NIH Office of the Director | 1 R01 AI130345 01 | Frédéric Geissmann |
| Ludwig Center for Cancer Immunotherapy | Basic and Translational Immunology Grants | Frédéric Geissmann |
| Cycle for Survival | | Frédéric Geissmann |
| American Association for Cancer Research | 18-40-15-VICA | Rocio Vicario |
| New York State Stem Cell Science | C32559GG | Leslie Weber |
| Charles H Revson Fellowship | | Leslie Weber |

The funders had no role in study design, data collection and interpretation, or the decision to submit the work for publication.

### Author contributions

Rocio Vicario, Conceptualization, Resources, Data curation, Formal analysis, Supervision, Funding acquisition, Validation, Investigation, Methodology, Writing – original draft, Project administration, Writing – review and editing; Stamatina Fragkogianni, Nicholas D Socci, Data curation, Formal analysis, Methodology; Leslie Weber, Data curation, Formal analysis, Investigation; Tomi Lazarov, Conceptualization, Data curation, Formal analysis, Investigation; Yang Hu, Samantha Y Hayashi, Barbara

Craddock, Masato Ogishi, Bertrand Boisson, Olivier Elemento, Jean-Laurent Casanova, Data curation, Formal analysis; Araitz Alberdi, Formal analysis, Project administration; Ann Baako, Oyku Ay, Ting Zhou, Methodology; Estibaliz Lopez-Rodrigo, W Todd Miller, Formal analysis, Investigation, Methodology; Rajya Kappagantula, Christine A Iacobuzio-Donahue, Netherlands Brain Bank, Resources; Agnes Viale, Resources, Methodology; Richard M Ransohoff, Richard Chesworth, Funding acquisition, Investigation; Omar Abdel-Wahab, Investigation; Frédéric Geissmann, Conceptualization, Resources, Data curation, Formal analysis, Supervision, Funding acquisition, Investigation, Methodology, Writing – original draft, Project administration, Writing – review and editing

**Author ORCIDs**
Rocio Vicario ⓘ https://orcid.org/0000-0002-7894-5261
Tomi Lazarov ⓘ https://orcid.org/0000-0002-6312-0080
Richard M Ransohoff ⓘ https://orcid.org/0000-0003-0175-6910
Omar Abdel-Wahab ⓘ https://orcid.org/0000-0002-3907-6171
Frédéric Geissmann ⓘ https://orcid.org/0000-0001-5029-2468

**Ethics**
Tissue samples. The study was conducted according to the Declaration of Helsinki. Human tissues were obtained with patient-informed consent and used under approval by the Institutional Review Boards from Memorial Sloan Kettering Cancer Center (IRB protocols #X19-027). Snap-frozen human brain and matched blood were provided by the Netherlands Brain Bank (NBB), the Human Brain Collection Core (HBCC, NIH), Hospital Sant Joan de Déu and the Rapid Autopsy Program (MSKCC, IRB #15-021). Samples were neuropathologically evaluated and classified by the collaborating institutions as Alzheimer's disease (AD) 1-5 or non-dementia controls. The mean age of AD patients is 65 years old (55.5% female, 44.5% male). The mean age of all controls is 54 years old (60% female, 40% male), and the mean age of AD age-matched controls was 70 years old (60% female , 40% male). The overall mean of the post-mortem delay interval was 9.8 hours. Patients did not present with germline pathogenic PSEN1/2/3 or APP AD's associated variants. For additional information on donor's brain regions, sex, age, cause of death, Apoe status, Braak status, see Supplementary file 1. To avoid possible contamination of sequencing data with mutations associated with donor's tumoral disease in the group of non-dementia controls, we refrained from selecting cases with blood malignances or with brain tumors. Samples from histiocytosis patients were collected under GENE HISTIO study (approved by CNIL and CPP Ile-de France) from Pitié-Salpêtrière Hospital and Hospital Trousseau and from Memorial Sloan Kettering Cancer Center.

Reviewer #1 (Public review): https://doi.org/10.7554/eLife.96519.3.sa1
Reviewer #2 (Public review): https://doi.org/10.7554/eLife.96519.3.sa2
Author response https://doi.org/10.7554/eLife.96519.3.sa3

---

# Additional files

**Supplementary files**
Supplementary file 1. Characteristics of Alzheimer's disease (AD) and control donors and samples.

Supplementary file 2. Targeted-sequencing gene panel.

Supplementary file 3. Variants identified in Alzheimer's disease and control brain samples.

Supplementary file 4. Pathway enrichment analysis for genes target of pathogenic variants in PU.1 samples.

Supplementary file 5. BRAFV600E in brain PU.1+ cells from Histiocytosis patients.

Supplementary file 6. Predicted deleterious variants by whole exome sequencing (WES).

Supplementary file 7. RNAseq analysis of mouse cell lines: Differential expressed genes and GSEA analysis.

Supplementary file 8. RNAseq analysis of human induced Pluripotent Stem Cells (hIPSC) derived microglial-like cells: Differential expressed genes and gene set enrichment analysis (GSEA).

Supplementary file 9. Single nuclei RNAseq analysis of control and Alzheimer's disease (AD) microglia: Differential expressed genes per clusters and gene set enrichment analysis (GSEA).

MDAR checklist

## Data availability

DNA sequencing data processed for selection of somatic variants are available for all patients and samples in *Supplementary file 3*. Raw DNA sequencing data (FASTQ files) from targeted-deep sequencing are deposited in dbGaP under project accession number phs002213.v1.p1, for samples where patient-informed consent for public deposition of DNA sequencing data was obtained. Bulk RNAseq raw data from IPSCs and MAC lines are deposited in GEO (GSE274037). Sn-RNAseq raw data are deposited in GEO (GSE286627) and as an interactive analysis web tool accessible at https://weillcornellmed.shinyapps.io/Human_brain/.

The following datasets were generated:

| Author(s) | Year | Dataset title | Dataset URL | Database and Identifier |
|---|---|---|---|---|
| Vicario R | 2025 | DNA sequencing of postmortem brain samples processed for selection of somatic variants | https://www.ncbi.nlm.nih.gov/projects/gap/cgi-bin/study.cgi?study_id=phs002213.v1.p1 | NCBI dbGaP, phs002213.v1.p1 |
| Geissmann F, Vicario R | 2025 | A microglia clonal inflammatory disorder in Alzheimer's Disease | https://www.ncbi.nlm.nih.gov/geo/query/acc.cgi?acc=GSE274037 | NCBI Gene Expression Omnibus, GSE274037 |
| Geissmann F, Vicario R | 2025 | A microglia clonal inflammatory disorder in Alzheimer's Disease II | https://www.ncbi.nlm.nih.gov/geo/query/acc.cgi?acc=GSE286627 | NCBI Gene Expression Omnibus, GSE286627 |

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
