## [Editor Report · eLife assessment]

This **fundamental** study enhances our understanding of how somatic variants in microglia might influence the onset and progression of neurodegenerative diseases such as Alzheimer's. The evidence supporting the conclusions is **compelling**, with the authors employing a multi-faceted approach to identify an enrichment of potentially pathogenic somatic mutations in Alzheimer's disease microglia. This research will be of significant interest to those investigating somatic mutations, Alzheimer's disease, microglial biology and cell signalling pathways.

---

## [Referee Report · Reviewer #1 (Public review)]

In the revised manuscript Vicario et al. provide new insights on a potential contribution of somatic mutations within the microglia population of the CNS that accelerates microglia activation and disease-associated gene signatures in Alzheimer's disease. Here they especially identified an "enrichment" of pathological SNVs in microglia, but not the peripheral blood, that are associated with clonal proliferative disorders and neurological diseases in a subset of patients with AD. They identified P-SNVs in microglia of AD patients located within the ring domain of CBL, a negative regulator of MAPK signaling. They further provide mechanistic insights how these variants result in MAPK over-activation and subsequently in a pro-inflammatory phenotype in human microglia-like cells in vitro.

Overall, this study provides novel evidence from an AD patient cohort pointing to a potential contribution of microglia-specific somatic mutations to disease onset and/or progression in at least a subset of patients with Alzheimer's disease.

The work within this study is highly relevant and will open new study lines to explore somatic mutations within the microglia compartment and neurodegenerative diseases.

Strengths:

As outlined above, the study identified P-SNVs in microglia of AD patients associated with clonal proliferative disorders, but also give an in depth analysis in re-occurring P-SNVs located within the ring domain of CBL, a negative regulator of MAPK signaling. They further provide mechanistic insights how these variants result in MAPK over-activation and subsequently in a pro-inflammatory phenotype in HEK cells, BV2 cells, MAC cells and human microglia-like cells in vitro. The over-activation of the cells in vitro is convincing.

Great care was taken to identify the limitations of the possible conclusions and to make careful conclusions. For example, they highlight that the pathway proposed to be affected may be an explanation for a subset of AD patients, and emphasize that it is yet unclear whether this accumulation of pathological SNVs is a cause or consequence of disease progression

The study supports an enrichment of P-SNVs in several genes associated clonal proliferative disorders in microglia and nicely separates this from SNVs associated with clonal hematopoiesis in the peripheral blood found in AD patients and controls.

The authors further acknowledged that several age matched control patients were diagnosed with cancer or tumor-associated diseases and carefully dissected the occurring SNVs in these patients are not associated with the P-SNVs identified in the microglial compartment of the AD cohort.

Weaknesses:

The revised study is overall convincing and has improved in the revised version, but some points especially regarding the clear connection of the seen somatic variants in microglia with a potential role in disease progression remain unanswered.

A potential connection between P-SNVs in microglia and disease pathology and symptoms was not further explored by the authors but might be in future work.

Taken this into account, maybe the title is a bit overstated and could be tuned down.

---

## [Referee Report · Reviewer #2 (Public review)]

Summary:

In this study, Vicaro et al. aimed to quantify and characterize mosaic mutations in human sporadic Alzheimer's disease (AD) brain samples. They focused on three broad classes of brain cells, neurons that express the marker NeuN, microglia that express the marker PU.1, and double-negative cells that presumably comprise all other brain cell types, including astrocytes, oligodendrocytes, oligodendrocyte progenitor cells, and endothelial cells. The authors find an enrichment of potentially pathogenic somatic mutations in AD microglia compared to controls, with MAPK pathway genes being particularly enriched for somatic mutations in those cells. The authors report a striking enrichment for mutations in the gene CBL and use in vitro functional assays to show that these mutations indeed induce MAPK pathway activation.

The current state of the AD and somatic mutation fields puts this work into context. First, AD is a devastating disease whose prevalence is only increasing as the population of the U.S. is aging, necessitating the investigation of novel features of AD to identify new therapeutic opportunities. Second, microglia have recently come into focus as important players in AD pathogenesis. Many AD risk genes are selectively expressed in microglia, and microglia from AD brain samples show a distinct transcriptional profile indicating an inflammatory phenotype. The authors' previous work shows that a genetic mouse model of mosaic BRAF activation in macrophages (including microglia) displays a neurodegenerative phenotype similar to AD (Mass et al., 2017, doi:10.1038/nature23672). Third, new technological developments have allowed for identifying mosaic mutations present in only a small fraction of or even single cells. Together, these data form a rationale for studying mosaic mutations in microglia in AD. In light of the authors' findings regarding MAPK pathway gene somatic mutations, it is also important to note that MAPK has previously been implicated in AD neuroinflammation in the literature.

Strengths:

The study demonstrated several strengths. Firstly, the authors used two methods to identify mosaic mutations: (1) deep (~1,100x) DNA sequencing of a targeted panel of >700 genes they hypothesized might, if mutated somatically, play a role in AD, and (2) deep (400x) whole-exome sequencing (WES) to identify clonal mosaics outside of those genes. A second strength is the agreement between these experiments, where WES found many variants identified in the panel experiment, and both experiments revealed somatic mutations in MAPK pathway genes. Third, the authors demonstrated in several in vitro systems that many mutations they identified in MAPK genes activate MAPK signaling. Finally, the authors showed that in some human brain samples, single-cell gene expression analysis revealed that cells bearing a mosaic MAPK pathway mutation displayed dysregulated inflammatory signaling and dysregulation in other pathways. This single-cell analysis was in agreement with their in vitro analyses.

Weaknesses:

The study also showed some weaknesses. The sample size (45 AD donors and 44 controls) is small, reflected in the relatively modest effect sizes and p-values observed. This weakness is partially ameliorated by the authors' extensive molecular and functional validation of mutation candidates. Secondly, as the authors point out, this study cannot conclude whether microglial mosaic mutations cause AD or are an effect of AD. Future studies may shed more light on this important question.

Conclusions and Impact:

Considering the study's aims, strengths, and weaknesses, I conclude that the authors achieved their goal of characterizing the role of mosaic mutations in human AD. Their data strongly suggest that mosaic MAPK mutations in microglia are associated with AD. The impacts of this study remain to be seen, but they could include attempts to target CBL or other mutated genes in the treatment of AD. This work also suggests a similar approach to identifying potentially causative somatic mutations in other neurodegenerative diseases.

---

## [Author Response]

The following is the authors’ response to the original reviews.

We edited the manuscript for clarity, added information described in new figure panels (below) and corrected typos.

In figure 1 we corrected a typo.

In figure 2, panel 2H, and Figure S2E, we included a new statistical analysis (mixed effect linear regression) to compare mutational burden in controls and AD patients.

In figure 3, and Figure S4B, we revised the western blots panels in Panel 3E,F, to improve presentation of controls and quantification.

we corrected typos.

In figure 5 we removed a panel (former 5D) which did not add useful information.

In Figure S1A we included information about sex and age from the control and patients analyzed. In Figure S2B, we added an analysis of the mutational burden in controls, distinguishing controls with and without cancer.

We modified Table S1 for completeness of information for all samples analyzed.

**Reviewer #1:**
Weaknesses:Even though the study is overall very convincing, several points could help to connect the seen somatic variants in microglia more with a potential role in disease progression. The connection of P-SNVs in the genes chosen from neurological disorders was not further highlighted by the authors.

All P-SNVs are reported in Table S3.

We observed only two P-SNVs within genes associated to neurological disorders (brain panel in Table S2). - SQSTM1 (p.P392L) was identified in blood but not in brain from the patient AD48A.

- OPTN was identified (p.Q467P) in PU.1 from control 25.

To highlight this point, we modified the first paragraph of the discussion as follow:

“We report here that microglia from a cohort of 45 AD patients with intermediate-onset sporadic AD (mean age 65 y.o) is enriched for clones carrying pathogenic/oncogenic variants in genes associated with clonal proliferative disorders (Supplementary Table 2) in comparison to 44 controls. Of note we did not observe microglia P-SNVs within genes reported to be associated with neurological disorders (Supplementary Table 2) in patients, and one such variant was identified in a control (Supplementary Table 3) “.

The authors show in snRNA-seq data that a disease-associated microglia state seems to be enriched in patients with somatic variants in the CBL ring domain, however, this analysis could be deepened. For example, how this knowledge may translate to patient benefits when the relevant cell populations appear concentrated in a single patient sample (Figure 5; AD52) is unclear; increasing the analyzed patient pool for Figure 5 and showcasing the presence of this microglia state of interest in a few more patients with driving mutations for CBL or other MAPK pathway associated mutations would lend their hypotheses further credibility.

We acknowledge this limitation, but we respectfully submit that the analysis was performed in 2 patients. AD 53 also show a MAPK-associated inflammatory signature in the microglia clusters associated with mutations.

We performed the analysis on all FACS-purified PU.1+ nuclei samples that passed QC for single nuclei RNAseq. It should be noted that this analysis is extremely difficult with current technologies because microglia nuclei need to be fixed for PU.1 staining and FACS purification and the clones are small (~1% of microglia).

A potential connection between P-SNVs in microglia and disease pathology and symptoms was not further explored by the authors.

At the population level, Braak/CERAD scores, the presence of Lewy bodies, amyloid angiopathy, tauopathy, or alpha synucleinopathy were not different between AD patients with or without pathogenic microglial clones (Figure S3 and Table S1). Of note, we studied here a homogenous population of AD patients.

At the tissue level, the roles of mutant microglia in plaques for example is being investigated, but we do not have results to present at this time.

A recent preprint (Huang et al., 2024) connected the occurrence of somatic variants in genes associated with clonal hematopoiesis in microglia in a large cohort of AD patients, this study is not further discussed or compared to the data in this manuscript.

This pre-print supports the high frequency of detection of oncogenic variants associated with clonal proliferative disorders, they hypothesize that the mutations may be associated with microglia, but they only check a few mutations in purified microglia. Most of the study is performed in whole brain tissue. It does not really bring new information as compared to other study we cite in the introduction (and to our manuscript).

**Reviewer #2 (Recommendations For The Authors):**
Suggestions for improved or additional experiments, data, or analyses:The authors can demonstrate that identified pathological SNVs from their AD cohort also lead to the activation of human microglia-like cells in vitro, but do not provide any data from histological examination of the patient cohort (e.g. accumulation at the plaque site, microglia distribution, and cell number). The study could be further supported by providing a histological examination of patients with and without P-SNVs to identify if microglia response to pathology, microglia accumulation, or phagocytic capacity are altered in these patients.

We performed IBA1 staining in brain samples from control and from AD patients, with or without microglial clones and microglia density was not different between patient with and without mutations. In addition, histological reports from the brain bank Braak/CERAD scores, Lewis bodies, amyloid angiopathy, tauopathy, or alpha synucleinopathy did not suggest differences between patient with and without mutations (Figure S3). These results are preliminary and further investigations are ongoing.

It would have been interesting to see if for example, transgenic AD mice with an introduced somatic mutation in microglia show an altered disease progression with alterations in amyloid pathology or cognition.

We agree with the reviewer. We performed an in vivo study with mice expressing a 5xFAD transgene, an inducible microglia Cx3cr1CreERt2 BrafLSL-V600E transgene, or both, and performed survival, behavioral (Y-Maze and Novel Object Recognition), and histological analyses for β-Amyloid, p-Tau and Iba1 staining.

Microgliosis was increased in the group with the 2 transgenes, however the phenotype associated with the expression of a BrafV600E allele in microglia (Mass et al Nature 2017) was strongly dominant over the phenotype of 5xFAD mice, which did not allow us to conclude on survival and behavioral analyses.

Other studies with different transgenes are in progress but we have no results yet to include in this revised manuscript.

To connect the somatic mutations in microglia better to a potential contribution in neurodegeneration or neurotoxicity, the authors could provide further details on how to demonstrate if human microglia-like cells respond differentially to amyloid or induce neurotoxicity in a co-culture or slice culture model.

These studies are undertaken in the laboratory, but unfortunately, we have no results as yet to include in this revised manuscript.

The number of samples analyzed for hippocampi, especially in the age-matched controls might be underpowered.

Unfortunately, despite our best efforts, we were not able to analyze more hippocampus from control individuals. To control for bias in sampling as well as to other potential bias in our analysis, we investigated the statistical analysis of the cohorts for inclusion of age as a criterion (age matched controls), inclusion of a random effect structure, and possible confounding factor such as sex, brain bank site, and samples’ anatomical location (see revised Methods and revised Fig. 2C, F, and H, and S2B).

We first tested whether the inclusion of age is appropriate in a fixed-effects linear regression using a generalized linear model (GLM) with gaussian distribution. Compared to the baseline model, the model with age had significantly low AIC (from -66.6 to -71.9, P = 0.0067 by chi-square test). Therefore, the inclusion of age as a fixed effect is appropriate. We next tested multiple structures of mixed-effects linear modeling. We used donors as random effects, while utilizing age, disease status (neurotypical control vs. AD), or both as fixed effects. Fitting was performed using the lme function implemented in the nlme package with the maximum likelihood (ML) method. The incorporation of age and disease status significantly improved overall model fitting. Both age and AD are associated with a significant increase in SNV burden in this model (P<1x10^-4 and P=1x10^-4, respectively, by likelihood ratio test). The model's total explanatory power is substantial (conditional R^2=0.48). We also asked if the addition of potential confounding factors to the model is justified. Three factors were tested via the two above-mentioned methods: sex, brain bank site, and the anatomical location of the samples. In all cases, the AIC increased, and the P values by likelihood ratio tests were higher than 0.99. Therefore, from a statistical standpoint, the inclusion of these potential confounding factors does not seem to improve overall model fitting.

Minor corrections to the text and figures:The authors made a great effort to analyze various samples from one individual donor. One can get a bit confused by the sentence that "an average of 2.5 brains samples were analyzed for each donor". Maybe the authors could highlight more in the first paragraph of the results section and in Figure 1A, that there are multiple samples ("technical replicates") from one individual patient across different brain regions used.

We removed the ‘2.5’ sentence and rewrote the paragraph for clarity. Samples information’s are now displayed in Table S1.

In the method section is a part included "Expression of target genes in microglia", it was very hard to allocate where these data from public data sets were actually used and for which analysis. Maybe the authors could clarify this again.

AU response: we apologize and corrected the paragraph in the methods (page 6) as follow: “ Expression of target genes in microglia. To evaluate the expression levels of the genes identified in this study as target of somatic variants, we consulted a publicly available database (https://www.proteinatlas.org/), and also plotted their expression as determined by RNAseq in 2 studies (Galatro et al. GSE99074 33, and Gosselin et al. 34) (Table S3 and Figure S2). For data from Galatro et al. (GSE99074) 33, normalized gene expression data and associated clinical information of isolated human microglia (N = 39) and whole brain (N = 16) from healthy controls were downloaded from GEO. For data from Gosselin et al. 34, raw gene expression ­data and associated clinical information of isolated microglia (N = 3) and whole brain (N = 1) from healthy controls were extracted from the original dataset. Raw counts were normalized using the DESeq2 package in R 35.”

Table S3 is very informative, but also very complex. The reader could maybe benefit a lot from this table if it can be structured a bit easier especially when it comes to identifying P-SNVs and in which tissue sample they were found and if this was the same patient. The sorting function on top of the columns helps, but the color coding is a bit unclear.

Despite our best efforts we agree that the table, which contain all sequencing data for all samples, is complex. The color coding (red) only highlights the presence of pathogenic mutation.

**Reviewer #3 (Recommendations For The Authors):**
This is a well-done study of an important problem. I present the following minor critiques:At the bottom of Page 4 and into the top of Page 5, the authors state that 66 of the 826 variants identified in their panel sequencing experiment were found in multiple donors. Then the authors proceed to analyze the remaining 760 variants. It seems that the authors concluded that these multi-donor mosaics were artifacts, which is why they were excluded from further analysis. I think this is a reasonable assumption, but it should be stated explicitly so it is clear to the reader. Complicating this assumption, however, the authors later state that one of their CBL variants was found in two donors, and it is treated as a true mosaic. The authors should make it clear whether recurrent variants were filtered out of any given analysis. It remains possible that all recurrent variants are true mosaics that occurred in multiple donors. The authors should do a bit more to characterize these recurrent variants. Are they observed in the human population using a database like gnomAD, which, together with their recurrence, would strongly suggest they are germline variants? Are they in MAPK genes, or otherwise relevant to the study?

We apologize for the confusion. Our original intent for the ddPCR validation of variants (Figure 1E) was to count only 1 ‘unique’ variant for variants found for example in 1 brain sample and in the blood from the same patient, or in 2 brain regions from one patient, in order to avoid the criticism of overinflating our validation rate. This was notably the case for TET2 and DNMT3 variants. For example, validation of a TET2 variant found in 2 different brain areas and blood of the same donor is counted as 1 and not 3. We did not eliminate these variants from the analysis as they passed the criteria for somatic variants as presented in Methods.

In contrast, when a specific variant was found and validated in two different donors, we counted it as 2.

The characterization of variants included multiple parameters and databases, including for example AF and gnomAD, as indicated in Methods and reported in Table S3.

All ddPCR results can be found at the end of Table S3.

Figure 2B labels age-matched controls as "C", but Figure 2C labels age-matched controls as AM-C. Labels should be consistent throughout the manuscript.

We corrected this in the revised version.

It is not clear if the "p:0.02" label in Figure 2F is referring to AM-C Cx vs. AD-Cx or AM-C vs. AD. Please clarify.

We apologize for the confusion, and we corrected the legend. The calculated p value is for the comparison between Cortex from Controls (age-matched) and the Cortex from AD.

On Page 7, the authors state, "The allelic frequencies at which MAPK activating variants are detected in brain samples from AD patients range from ~1-6% of microglia (Fig. 3G), which correspond to clones representing 2 to 12% of mutant microglia in these samples, assuming heterozygosity." I understand what the authors mean here but I think it's a bit confusingly stated. I suggest something like "The allelic frequencies at which MAPK activating variants are detected in brain samples from AD patients range from ~1-6% in microglia (Figure 3G), which correspond to mutant clones representing 2 to 12% of all microglia in these samples, assuming heterozygosity."

We thank the reviewer for this suggestion and re-wrote that sentence.

Is there any evidence that the transcriptional regulators mutated in AD microglia (MED12, SETD2, MLL3, DNMT3A, ASXL1, etc.) are involved in regulating MAPK genes? This would tie these mutations into the broader conclusions of the paper.

This is a very interesting question, and indeed published studies indicate that some of the transcriptional /epigenetic regulators regulate expression of MAPK genes. However, in the absence of experimental evidence in microglia and patients, the argument may be too speculative to be included.

Do the authors have any thoughts as to whether germline variants in CBL are linked to AD? If not, why do they think germline mutations in CBL are not relevant to AD?

This is also a very interesting question. As indicated in our manuscript, germline mutations in CBL (and other member of the classical MAPK genes, see Figure 3C) cause early onset (pediatric) and severe developmental diseases known as RASopathies, characterized by multiple developmental defects, and associated with frequent neurological and cognitive deficits.

It is possible that some other (and more frequent?) germline variants may be associated with a late-onset brain restricted phenotype, but we did not find germline pSNV in our patients. GWAS studies may be more appropriate to test this hypothesis.

Do any donors show multiple variants? I don't think this is addressed in the text.

We do find donors with multiple variants (see Figure 3D and Figure S3), however at this stage, we did not perform single nuclei genotyping to investigate whether they are part of the same clone.

Figure S3 appears to be upside down.

This was corrected

Figure 5C should have some kind of label telling the reader what gene set is being depicted.

We added this information above the panel (it was in the corresponding legend).

At the top of Page 12, Lewy bodies are written as Lewis bodies.

This was corrected

Many control donors died of cancer (Table S1). Is there any information on which, if any, chemotherapeutics or radiation these patients received? Might this impact the somatic mutation burden? The authors should compare controls with and without cancer or with and without cancer treatments to rule this out.

As suggested by the reviewer, we analyzed the mutational load of age-matched controls with and without cancer (revised Figure S2B). As expected, we saw an increase in the mutational load in controls with cancer, particularly in their blood. This information was added in the result section.

This is most likely associated with the treatments received as well as possible cancer clones.

The formatting for Table S3 is odd. Multiple different fonts are used (this is also seen in Table S5). Column Q has no column ID. The word "panel" is spelled "pannel." The word "expressed" is spelled "expressd" in one of the worksheet labels. Columns BG-BN in the ALL-SNV worksheet are blank but seemingly part of the table.

We fixed this error in Table S3.